# *RetrievalAttention*: Accelerating Long-Context LLM Inference via Vector Retrieval

## Abstract

Transformer-based Large Language Models (LLMs) have become increasingly important. However, due to the quadratic time complexity of attention computation, scaling LLMs to longer contexts incurs extremely slow inference speed and high GPU memory consumption for caching key-value (KV) vectors. This paper proposes **RetrievalAttention**, a training-free approach to both accelerate attention computation and reduce GPU memory consumption. By leveraging the dynamic sparsity of attention mechanism, RetrievalAttention proposes to build approximate nearest neighbor search (ANNS) indexes for KV vectors in CPU memory and retrieve the most relevant ones through vector search during generation. Unfortunately, we observe that the off-the-shelf ANNS indexes are often ineffective for such retrieval tasks due to the out-of-distribution (OOD) between query vectors and key vectors in the attention mechanism. RetrievalAttention addresses the OOD challenge by designing an attention-aware vector search algorithm that can adapt to the distribution of query vectors. Our evaluation demonstrates that RetrievalAttention achieves near full attention accuracy while only requiring access to 1–3% of the data. This leads to a significant reduction in the inference cost of long-context LLMs, with a much lower GPU memory footprint. In particular, RetrievalAttention only needs a single NVIDIA RTX4090 (24GB) to serve 128K tokens for LLMs with 8B parameters, which is capable of generating one token in 0.188 seconds.

## 1 Introduction

Recent transformer-based Large Language Models (Vaswani et al., 2017) have shown remarkable capabilities in processing long contexts. For instance, Gemini 1.5 Pro (Team, 2024) has supported the context window of up to 10 million tokens. While this is promising for analyzing extensive data, supporting longer context windows also introduces challenges for inference efficiency due to the quadratic complexity of attention computation. To enhance efficiency, KV caching, a technique that retains past key and value vectors, has been widely adopted to prevent redundant computations. However, KV caching-based systems face two primary issues: (a) substantial GPU memory requirements, particularly for long contexts, e.g., the Llama-3-8B model requires approximately 125GB per million tokens; and (b) inference latency increases linearly to the context size, primarily due to the time needed to access cached tokens — a common issue across various computing devices, including GPUs. Therefore, reducing storage costs and token access is vital for enhancing inference efficiency.

The solution lies in leveraging the dynamic sparsity inherent in the attention mechanism (Deng et al., 2024). This refers to the phenomenon where each query vector significantly interacts with only a limited subset of key and value vectors, with the selection of these critical vectors varying dynamically based on individual queries. Prior work (Tang et al., 2024; Xiao et al., 2024a; Ribar et al., 2024; Lee et al., 2024; Singhania et al., 2024) has proposed various techniques to capitalize on this observation to improve the efficiency of attention computation. However, most of these methods identify important tokens either statically (Xiao et al., 2024b; Li et al., 2024) or heuristically (Xiao et al., 2024a; Ribar et al., 2024; Tang et al., 2024), leading to imprecise approximations that often result in a significant performance drop.

We observe that the Approximate Nearest Neighbor Search (ANNS) index, such as proximity graph (Malkov & Yashunin, 2018), is particularly effective in this context. ANNS index is used to efficiently find the most similar vectors to the query and is widely adopted in various domains

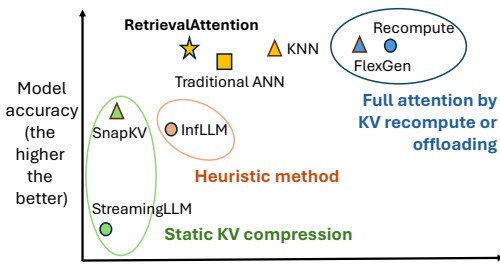

Figure 1: RetrievalAttention achieves similar task accuracy as full attention but exhibits extremely low decoding latency.

Table 1: Decoding latency and memory required for KV cache of Llama-3-8B across different context lengths on one A100 GPU.

| Prompt Length | 128K | 256K | 512K | 1M |
|---|---|---|---|---|
| Total Latency (s) | 32.8 | 111 | 465 | 1,765 |
| FFN (s) | 7.6 | 15 | 31 | 70 |
| Attention (s) | 25.2 | 96 | 434 | 1,695 |
| GPU Memory KV Cache (GB) | 15.6 | 31.2 | 62.5 | 125 |

like information retrieval (Xiong et al., 2021) and recommendation systems (Cost & Salzberg, 1993; Covington et al., 2016; Pal et al., 2020). When using the inner product as the similarity measurement to build the index for key vectors, searching over the index with the query vector exactly aligns with the attention mechanism.[1] It can directly identify the most critical key vectors with the maximum inner product to the query vector in sub-linear time complexity, yielding a higher accuracy compared to previous static or heuristic methods (as illustrated in Figure 1). Furthermore, most ANNS algorithms are compatible with CPU implementation, which enables strategic allocation of GPU and CPU memory resources and thus facilitates the handling of longer context inference on devices with limited GPU memory.

Leveraging ANNS for attention mechanism presents a unique challenge: the out-of-distribution (OOD) problem between query and key vectors. Most ANNS engines operate under the assumption that both query and key vectors are drawn from the same data distribution. However, this assumption does not hold in this context due to the different projection weights for query and key vectors in attention mechanism. The Mahalanobis distance (Mahalanobis, 2018) shows that query vectors deviate more than $10\times$ farther from key vectors compared to that between in-distribution query and key vectors. Unfortunately, the effectiveness of ANNS degrades significantly under OOD problem. In particular, our empirical analysis indicates that maintaining an acceptable level of inference accuracy requires conventional ANNS scanning 30–50% of all key vectors to identify the critical ones, which fails to fully leverage the inherent sparsity of the attention mechanism and impairs the inference latency. To the best of our knowledge, we are the first to identify the challenge of OOD in using ANNS index for attention computation, a factor that is crucial for inference efficiency and accuracy.

In this work, we present RetrievalAttention, an efficient method for accelerating long-context LLM generation. RetrievalAttention employs dynamic sparse attention during token generation, allowing the most critical tokens to emerge from the extensive context data. To address the challenge of OOD, RetrievalAttention proposes a vector index tailored for the attention mechanism, focusing on the distribution of queries rather than keys. This approach allows for the traversal of only a small subset of key vectors (1–3%) to identify the most relevant tokens, yielding accurate attention scores and inference accuracy. In addition, RetrievalAttention reduces GPU memory consumption by retaining a small number of KV vectors in GPU memory following static patterns (e.g., similar to StreamingLLM (Xiao et al., 2024b)) and offloading the majority of KV vectors to CPU memory for index construction. During token generation, RetrievalAttention efficiently retrieves critical tokens using ANNS index on the CPU and merges the partial attention results from both the CPU and GPU. This strategy enables RetrievalAttention to perform attention computation with reduced latency and minimal GPU memory footprint.

We evaluate the accuracy and efficiency of RetrievalAttention on both commodity GPUs (RTX4090) and high-end GPUs (A100) on three long-context LLMs across various long-context benchmarks like ∞-Bench (Zhang et al., 2024b) and RULER (Hsieh et al., 2024). For the 128K context on the RTX4090 GPU, RetrievalAttention achieves $4.9\times$ and $1.98\times$ decoding-latency reduction compared to the retrieval method based on exact KNN and traditional ANNS index, respectively, while maintaining

---

[1]Maximum inner product search can be viewed as similarity search and efficiently solved by ANNS indexes (Morozov & Babenko, 2018).

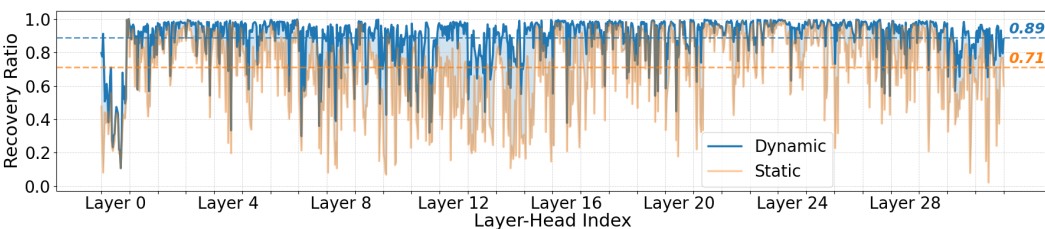

Figure 2: The dynamic sparsity of each layer and head in Llama-3-8B model in the KV retrieval test of 100,000 tokens. The blue curve shows that using dynamically selected top-1000 critical tokens achieves an average recovery ratio of 89%, indicating high attention sparsity. In contrast, the orange curve reveals that statically using the initially determined top-1000 critical tokens from the generation of the first token to generate subsequent tokens drops the average recovery ratio to 71%.

the same accuracy as full attention. To the best of our knowledge, RetrievalAttention is the first solution that supports running 8B-level models on a single RTX4090 GPU (24GB) with acceptable latency and almost no accuracy degradation.

## 2 BACKGROUND AND MOTIVATION

### 2.1 LLM AND ATTENTION OPERATION

In the generation process of the $t$-th token, the attention operation computes the dot product between the query vector $\mathbf{q}_t \in \mathbb{R}^{1 \times d}$ (where $d$ is the hidden dimension) and the key vectors of all preceding tokens $\mathbf{k}_i \in \mathbb{R}^{1 \times d}$ (for $i \leq t$). This product is scaled by $d^{-\frac{1}{2}}$ and normalized via a `Softmax` function to yield the attention score $a_{t,i}$. These scores then weight the values $\mathbf{v}_i$, resulting in the output $\mathbf{o}_t$.

$$z_i = \frac{\mathbf{q}_t \cdot \mathbf{k}_i^T}{\sqrt{d}}, \quad a_{t,i} = \frac{e^{z_i}}{\sum_{j=1..t} e^{z_j}}, \quad \mathbf{o}_t = \sum_{i=1..t} a_{t,i} \cdot \mathbf{v}_i \quad (1)$$

LLM inference contains two stages: the prefill phase and decoding phase. The prefill phase, which only happens once, takes all tokens of the prompt as input and performs attention with a time-complexity $O(n^2)$. In the decoding (token generation) phase, the newly generated token is added to the input and computes attention scores with same complexity. One common optimization to avoid repetitive calculation is caching past KV states, thereby reducing the complexity to $O(n)$.

### 2.2 EXPENSIVE LONG-CONTEXT SERVING

Due to the quadratic time complexity of attention operation, serving long-sequence input incurs extremely high costs. Table 1 shows the inference latency of Llama-3-8B without KV cache. When the prompt length reaches 1 million tokens, generating every token requires 1,765 seconds with over 96% of latency spent on attention operations. Although KV cache can reduce the decoding latency, it demands a huge amount of GPU memory for long contexts. As shown in Table 1, 125 GB memory is necessary for storing the KV cache when the context length reaches 1 million tokens, which is far beyond the GPU memory capacity of commodity GPUs such as the RTX4090 (24GB) or even high-end GPUs like A100 (40GB or 80GB). This necessitates either scaling to more GPUs to accommodate the large KV cache (Liu et al., 2024a) or repetitively offloading and reloading the entire KV cache between CPU and GPU memory over PCIe (Sheng et al., 2023), resulting in excessive communication overhead. Neither approach provides an efficient and cost-effective solution for long-context inference on commodity GPUs.

### 2.3 DYNAMIC AND SPARSE ATTENTION

Corroborating recent work (Xiao et al., 2024b; Li et al., 2024), we observe that attention computation in LLMs exhibits significant sparsity. Despite the large context length, only a small fraction of tokens

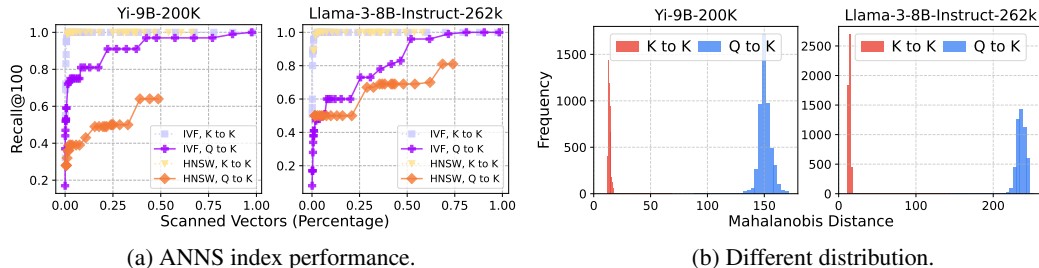

(a) ANNS index performance.                    (b) Different distribution.

Figure 3: (a) Query vectors ($Q$) and key vectors ($K$) are dumped from Yi-9B and Llama-3-8B with a prompt length of 128,000 tokens. Off-the-shelf ANNS indexes perform poorly on $Q$ to $K$ searches while work well for $K$ to $K$ searches. (b) Query vectors are distant from key vectors, while key vectors themselves are close.

with the highest attention scores (i.e., $a_{t,i}$ in Equation 1), also known as critical tokens, contribute significantly to the attention output.

We quantify the attention sparsity by calculating the cumulative sum of attention scores of top-$k$ critical tokens. This cumulative sum, called recovery ratio, represents how much of the full attention can be recovered using a small number of critical tokens, with a higher recovery ratio indicating greater sparsity. When generating 20 tokens consecutively based on a prompt of 100,000 tokens, we profile the average recovery ratio of decoding tokens using top-1000 critical tokens in different layers and heads of the model. As shown in the blue curve of Figure 2, by accurately selecting top-1000 critical tokens based on full attention, most attention heads can recover over 90% of the attention scores from the full attention, with an average of 89% across all heads and layers.

Furthermore, we observe that as the LLM continues generating new tokens, the critical key vectors change dynamically, highly depending on the current query vector. To verify this, we first collect the top-1000 critical key vectors to generate the first token in each attention head and statically apply them for the subsequent token generation. The results shown in the orange curve of Figure 2 indicate a significant drop in the average recovery rate, from 89% to 71%. This demonstrates that tokens considered important in previous queries may not be critical in subsequent queries, and vice versa. Therefore, it is necessary to dynamically select important tokens for each query vector.

The dynamic sparsity shows a promising path to approximately compute attention with greatly reduced cost and without sacrificing the model accuracy. For each query, if we can accurately identify the relevant key-value vectors with higher importance, minimum GPU memory and a much lower time complexity can be achieved for attention computation.

## 2.4 CHALLENGES OF OFF-THE-SHELF VECTOR SEARCH

To reduce the latency of long contexts inference while maintaining performance, we require a method to accurately identify the critical tokens to the current query in sub-linear time complexity. Additionally, given the constrained GPU memory, it would be beneficial if such a method could efficiently utilize CPU memory to manage the KV vectors. Based on Equation 1, one key vector is critical for a query vector if they have a large inner product. With the inner product as a similarity function, performing searches on ANNS indexes aligns well with the goal of the attention mechanism to efficiently find critical key vectors for a query vector.

Traditional ANNS indexes generally cluster similar (close) vectors and select the representative vector for each cluster (Sivic & Zisserman, 2003) or directly build connections between similar vectors to form a proximity graph (Wang et al.).[2] For cluster-based indexes, the query first compares with all representative vectors and then only accesses the most similar clusters, whereas, in the proximity graph, the query performs a greedy search, moving closer to the most similar vectors at each hop. Both methods typically require scanning a limited subset of all vectors (e.g., 1%) to identify the most similar vectors to the query, achieving high search efficiency and accuracy. However, we find that

---

[2]In this context, we use "similar" and "close" to indicate vectors with larger inner product interchangeably.

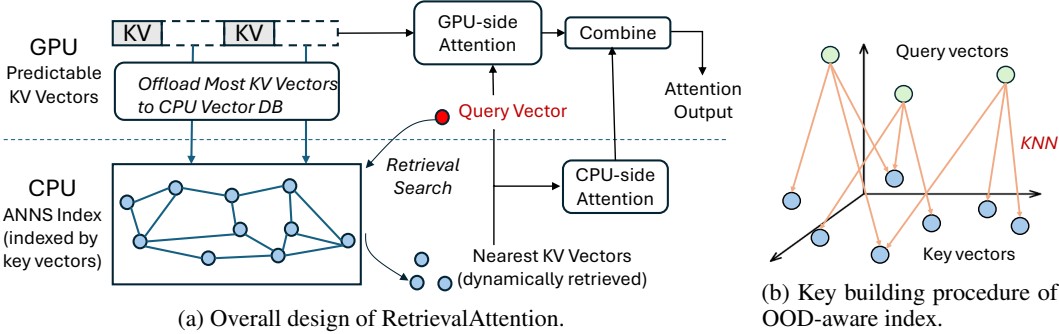

(a) Overall design of RetrievalAttention.

(b) Key building procedure of OOD-aware index.

Figure 4: (a) RetrievalAttention offloads most KV tokens to vector databases in CPU, which are retrieved during the decoding phase to find the most relevant KV tokens with queries. (b) During the index construction, we link each query to its exact top-$k$ nearest key vectors (KNN).

naively applying off-the-shelf vector indexes fails to provide good performance because of the OOD issue between query ($Q$) and key vectors ($K$).

In conventional vector databases, the distribution of vectors between content and query is often well-aligned because they are derived from the same embedding model. However, naively using traditional vector indexes for attention computation suffers from an inherent distribution gap between queries and keys, which are projected by different weights as 2.1. Figure 3a (focus on $Q$ to $K$ for now) demonstrates the performance of widely-used vector indexes supported by Faiss (Douze et al., 2024) using a query vector to retrieve the most similar key vectors. It compares the percentage of keys scanned and the corresponding recall achieved (i.e., the overlapping ratio between the retrieved top-100 results and the ground truth). Cluster-based IVF (Sivic & Zisserman, 2003) requires scanning $\sim$30–50% data for a recall rate higher than 0.95, and graph-based HNSW (Malkov & Yashunin, 2018) falls into a local optimum. The results show that traditional vector indexes require scanning a large number of vectors to achieve a high recall, highlighting the challenge of performing efficient vector searches for attention.

Fundamentally, the difficulty is due to the OOD between query and key vectors. We quantify this using Mahanobis distance (Mahalanobis, 2018), which measures the distance from a vector to a distribution. We sample 5,000 vectors from $Q$ and $K$ respectively as the query set and compute the the Mahanobis distance from the query set to the remaining vectors in $K$. Figure 3b shows that the queries from $Q$ are significantly distant from the $K$ vectors (OOD) while $K$ themselves are very close. Therefore, traditional index building based solely on the closeness between key vectors does not align with the attention mechanism, which requires to retrieve critical tokens as nearest neighbors from the query vectors' viewpoint. In contrast, Figure 3a shows that using sampled $K$ as the queries ($K$ to $K$) can easily achieve a high recall by only scanning 1–5% vectors because they are in the same distribution. Similarly, query vectors in each attention head also follow the same distribution as they are projected by the same model weight. For efficient vector search, the index must consider the OOD characteristic of the attention computation by design.

## 3 RETRIEVALATTENTION DESIGN

In this work, we focus on the acceleration of token generation and assume the prefill of the long-context prompt is done in advance, which is widely supported by existing LLM service providers (e.g., context caching (Google Cloud, 2024) or separation of prefill and decoding (Patel et al., 2024; Qin et al., 2024)).

We propose RetrievalAttention that leverages attention-aware vector search to approximate attention computation by CPU-GPU co-execution accurately. Figure 4a shows the overall design of RetrievalAttention. Based on our observation in §2.3, We derive an approximated attention by selectively retrieving relevant key-value vectors while discarding those that are negligible(§3.1). To efficiently support long context, we offload most KV vectors to the CPU memory, build vector indexes, and use attention-aware vector search to find critical tokens. (§3.2). To better exploit the

GPU devices, we leverage the attention scores obtained in the prefill phase to select a proportion of KV cache that is consistently important during the decoding phase and persist them on GPU devices. RetrievalAttention computes partial attention with dynamically retrieved from CPU memory and persistent key-value vectors in GPU memory in parallel and finally combines them together(§3.3).

## 3.1 Approximated Attention

Based on the Equation 1, RetrievalAttention approximates the full attention output $\mathbf{o}_t$ by selectively utilizing the KV vectors associated with high attention scores (*i.e.*, $a_{t,i}$). Specifically, we define $\mathcal{I}_{t,\epsilon}$ as a subset of token indices for which the attention score surpasses $\epsilon$. Consequently, a sparse attention mechanism, which only considers tokens located in $\mathcal{I}_{t,\epsilon}$, can be defined as follows:

$$\mathbf{o}_t = \sum_{i \in \mathcal{I}_{t,\epsilon}} a_{t,i} \cdot \mathbf{v}_i + \sum_{i \notin \mathcal{I}_{t,\epsilon}} \cancel{a_{t,i} \cdot \mathbf{v}_i} \approx \sum_{i \in \mathcal{I}_{t,\epsilon}} \tilde{a}_{t,i} \cdot \mathbf{v}_i \quad \text{where} \quad \tilde{a}_{t,i} = \frac{e^{z_i}}{\sum_{j \in \mathcal{I}_{t,\epsilon}} e^{z_j}} \tag{2}$$

Based on the above approximation, we build RetrievalAttention to only consider important key-value vectors (i.e., $\mathcal{I}_{t,\epsilon}$) that are persistent in GPU cache and dynamically retrieved by vector indexes.

## 3.2 Attention-aware Vector Search

For each pair of key and value vectors, we first decide whether to hold them in CPU or GPU memory (the decision method is elaborated in §3.3). The KV vectors offloaded to CPU memory will be indexed by $\mathbf{k}_i \in \mathcal{R}^d$ and queried by $\mathbf{q}_t$ to find the most relevant ones.

To accelerate the vector search during token generation, RetrievalAttention diverges from traditional indexes that only consider the closeness between key vectors for index building. Instead, it leverages the existing query vectors in the prefill phase to guide the index building for key vectors, efficiently mitigating the distribution gap. As shown in Figure 4b, during the index construction, RetrievalAttention explicitly establishes connections from the query vector to its nearest key vectors (i.e., exact $k$-nearest neighbors, or KNN). The KNN results can be efficiently computed via GPU, forming a mapping from query vector distribution to key vector distribution. Using this structure, the decoding query vector can first search its nearest query vectors and then obtain the most relevant key vectors through the distribution mapping.

Therefore, the KNN connections from query vectors to key vectors serve as a bridge to reconcile their distribution differences. However, this structure still has imperfections in both memory overhead and search efficiency because we need to store and access query vectors besides key vectors. To address this problem, we leverage the projection technique from the state-of-the-art cross-modal ANNS index RoarGraph (Chen et al., 2024a) to eliminate all query vectors. Specifically, we project the KNN connections into key vectors by linking key vectors that are connected to the same query vectors, which efficiently streamlines the search. This process connects key vectors that are perceived as close from the query vectors' perspective, allowing efficient index traversal for future query vectors.

Our evaluation shows that, by effectively modeling the proximity relationship between the query and key vectors, the vector database only requires scanning 1–3% key vectors to reach a high recall, significantly reducing the index search latency by 74% compared with IVF indexes (Sivic & Zisserman, 2003).

## 3.3 CPU-GPU Co-Execution

To exploit GPU parallelism and accelerate attention computation, RetrievalAttention decomposes the attention computation into two disjoint sets of KV cache vectors, the predictable ones on GPU and the dynamic ones on CPU, and then combines the partial attention outputs together.

We leverage the patterns observed in the prefill phase to predict KV vectors that are consistently activated during token generation. Similar to StreamingLLM (Xiao et al., 2024b), our current implementation uses fixed initial tokens and the last sliding window of the context as the static pattern and persists them in the GPU cache. RetrievalAttention can be adapted to utilize more complex static patterns (Li et al., 2024; Jiang et al., 2024), achieving the best trade-off between low inference cost and high accuracy. During the prefill phase, we physically separate the static tokens in the GPU

memory with the remaining tokens, which are offloaded to the CPU memory indexed by the ANNS. To minimize data transfer over the slow PCIe interface, RetrievalAttention independently computes the attention results for the CPU and GPU components and then combines them, inspired by the FastAttention (Dao et al., 2022).

## 4 EVALUATION

In this section, we compare the performance of RetrievalAttention in long-context LLM inference against full attention and other state-of-the-art methods. Through experiments, we mainly explore the following questions: (1) **How does RetrievalAttention affect the model's inference accuracy?** Specifically, we assess the generation accuracy of RetrievalAttention and other methods across various downstream tasks (§4.2) (2) **Can RetrievalAttention efficiently reduce the token generation latency of long-context LLM inference?** We compare the end-to-end decoding latency of RetrievalAttention with that of other baselines (§4.3).

### 4.1 EXPERIMENTAL SETUP

**Testbed, Models, and Configurations.** We conduct experiments on a server equipped with one NVIDIA RTX4090 GPU (24GB memory) and an Intel i9-10900X CPU with 10 physical cores (20 logical cores) and 128GB DRAM. The experiment results using NVIDIA A100 GPU can be found in §A.4. We implement RetrievalAttention on three state-of-the-art long-context LLMs, including Llama-3-8B-Instruct-262k (Gradient AI, 2024), Yi-6B-200K (01-ai, 2024a), and Yi-9B-200K (01-ai, 2024b). To show a practical speedup of RetrievalAttention and ensure the CPU memory consumption in long contexts does not exceed the DRAM capacity, we follow previous work (Tang et al., 2024) to run the benchmark in real-world single-batch scenarios.

**Baselines.** We compare RetrievalAttention with the following training-free baselines. (1) Full attention without KV cache as well as the version with KV cache using vLLM (Kwon et al., 2023). (2) StreamingLLM (Xiao et al., 2024b): it retains initial tokens along with fixed-length recent tokens in the GPU memory and discards remaining tokens. (3) SnapKV (Li et al., 2024): it only caches the critical tokens observed from the last window of the prompt. (4) InfLLM (Xiao et al., 2024a): it separates the KV cache of continuous token sequences into blocks and selects representative vectors for each block. In the decoding phase, the current query scans all representative vectors and retrieves top-$k$ blocks with the highest similarity. (5) Quest (Tang et al., 2024): it keeps track of the minimal and maximal key values in KV cache pages and estimates the criticality of a page using the query vector. (6) InfiniGen (Lee et al., 2024): it prefetches only the essential KV cache entries by speculating important tokens required for subsequent attention layers.

To better assess the effectiveness of our method, we introduce two additional baselines using traditional vector search methods from Faiss (Douze et al., 2024). Specifically, Flat is an exact KNN method that performs a linear scan of all key-value vectors, whereas IVF indexes key vectors through clustering. By default, all indexing-based methods retrieve the top-100 nearest key vectors.

**Benchmarks.** We adopt three representative long-context benchmarks for evaluation.

- $\infty$-Bench (Zhang et al., 2024b): this benchmark consists of 7 tasks, including three retrieval tasks (passKey retrieval, number retrieval, KV retrieval) and four realistic tasks (code debugging, math find, dialogue and multiple-choices questions). The average context length of $\infty$-Bench is over 100K tokens.
- RULER (Hsieh et al., 2024): a comprehensive long-context benchmark consisting of 4 categories and 13 tasks, including retrieval, multi-hop tracing, aggregation, and QA tasks. The prompt length ranges from 4K to 128K, allowing us to determine the actual context window size of models.
- Needle-in-a-haystack (Greg Kamradt, 2023): it challenges the models to accurately retrieve information (the "needle") hidden within a lengthy document (the "haystack").

### 4.2 ACCURACY ON LONG CONTEXT TASKS

$\infty$-**Bench.** As shown in Table 2, RetrievalAttention achieves comparable accuracy to the full attention, benefiting from its efficient dynamic retrieval of important tokens. Static methods, such

Table 2: Performance (%) of different methods and models on ∞-Bench. The size of the static pattern is consistently 640 (128 initial tokens + 512 tokens in the local window). All indexing-based methods, including Flat, IVF, and RetrievalAttention retrieve top-100 key vectors by default. In the relatively complicated task KV Retrieval, we include the results of retrieving top-2000 key vectors.

| | Methods | Act. Tokens | Retr.N | Retr.P | Retr.KV | Code.D | Math.F | En.QA | En.MC | Avg. |
|---|---|---|---|---|---|---|---|---|---|---|
| Llama-3-8B | FullAttention | 128K | 100.0 | 100.0 | 17.5 | 19.0 | 39.5 | 9.1 | 68.0 | 50.4 |
| | StreamingLLM | 2K | 5.0 | 5.0 | 1.0 | 18.5 | 40.0 | 6.0 | 66.0 | 20.2 (-30.2) |
| | SnapKV | 2K | 100.0 | 100.0 | 0.5 | 18.0 | 40.0 | 11.8 | 67.0 | 48.2 (-2.2) |
| | InfLLM | 640+2K | 100.0 | 100.0 | 0.5 | 20.5 | 48.0 | 7.0 | 37.0 | 44.7 (-5.7) |
| | InfiniGen | 2K | 99.5 | 100.0 | 0.0 | 17.5 | 39.0 | 7.3 | 57.5 | 45.8 (-4.6) |
| | Quest | 2K | 100.0 | 100.0 | 0.0 | 18.0 | 40.0 | 8.2 | 67.0 | 47.6 (-2.8) |
| | Flat | 640+100/2K | 100.0 | 100.0 | 8.5/14.5 | 19.0 | 40.0 | 7.5 | 67.0 | 48.9 (-1.5) / 49.7 (-0.7) |
| | IVF | 640+100/2K | 94.0 | 100.0 | 9.5/14.0 | 19.0 | 40.0 | 7.8 | 67.0 | 48.2 (-2.2) / 48.8 (-1.6) |
| | **RetrievalAttention** | 640+100/2K | 100.0 | 100.0 | 9.0/14.0 | 19.0 | 40.0 | 7.5 | 67.0 | **48.9 (-1.5) / 49.6 (-0.8)** |
| Yi-9B | FullAttention | 128K | 100.0 | 100.0 | 30.5 | 25.5 | 36.5 | 9.8 | 67.0 | 52.8 |
| | StreamingLLM | 2K | 5.0 | 5.0 | 0.5 | 24.0 | 33.5 | 6.4 | 72.0 | 20.9 (-31.9) |
| | SnapKV | 2K | 63.0 | 100.0 | 0.5 | 23.0 | 33.0 | 10.3 | 68.5 | 42.6 (-10.2) |
| | InfLLM | 640+2K | 100.0 | 100.0 | 0.5 | 20.5 | 43.0 | 9.4 | 44.0 | 45.3 (-7.5) |
| | Quest | 2K | 99.0 | 100.0 | 0.0 | 22.5 | 34.5 | 10.4 | 68.5 | 47.8 (-5.0) |
| | Flat | 640+100/2K | 100.0 | 100.0 | 21.0/30.0 | 23.0 | 35.0 | 10.8 | 68.5 | 51.2 (-1.6) / 52.5 (-0.3) |
| | IVF | 640+100/2K | 99.0 | 100.0 | 19.5/29.5 | 23.0 | 35.0 | 10.7 | 69.0 | 50.9 (-1.9) / 52.3 (-0.5) |
| | **RetrievalAttention** | 640+100/2K | 99.5 | 100.0 | 20.0/30.0 | 23.0 | 35.0 | 9.5 | 68.5 | **50.8 (-2.0) / 52.2(-0.6)** |
| Yi-6B | FullAttention | 128K | 98.0 | 100.0 | 3.5 | 31.0 | 11.0 | 19.2 | 55.5 | 45.5 |
| | StreamingLLM | 2K | 5.0 | 5.0 | 0.5 | 27.5 | 11.0 | 12.2 | 54.0 | 16.5 (-29.0) |
| | SnapKV | 2K | 39.0 | 100.0 | 0.0 | 30.5 | 8.5 | 17.1 | 55.0 | 35.7 (-9.8) |
| | InfLLM | 640+2K | 99.0 | 100.0 | 0.5 | 27.5 | 18.0 | 12.7 | 40.5 | 42.6 (-2.9) |
| | Quest | 2K | 98.5 | 100.0 | 0.0 | 30.5 | 8.5 | 17.3 | 54.5 | 44.2 (-1.3) |
| | Flat | 640+100/2K | 98.5 | 100.0 | 2.5/3.0 | 30.5 | 16.0 | 17.7 | 54.5 | 45.7 (+0.2) / 45.7 (+0.2) |
| | IVF | 640+100/2K | 98.0 | 100.0 | 2.5/3.5 | 29.5 | 16.0 | 17.5 | 54.5 | 45.4 (-0.1) / 45.6 (+0.1) |
| | **RetrievalAttention** | 640+100/2K | 95.0 | 99.0 | 3.0/3.0 | 30.0 | 16.0 | 17.6 | 54.5 | **45.0 (-0.5) / 45.0 (-0.5)** |

as StreamingLLM and SnapKV, lack this capability and, therefore, achieve sub-optimal accuracy. During token generation, the critical tokens change dynamically according to the current query, invalidating the previously captured static patterns. InfiniGen exhibits a noticeable drop in model accuracy compared to full attention due to inaccurate speculation of important tokens from previous layers. Although InfLLM and Quest supports dynamic retrieval of relevant blocks, it achieves nearly zero accuracy in complex tasks (i.e., KV retrieval) due to the low accuracy of representative vectors. Since RetrievalAttention can accurately identify the most relevant key vectors, it achieves the best accuracy in KV retrieval. Moreover, by retrieving more tokens (i.e., top-2000 shown in the column of Retr.KV) in KV retrieval, RetrievalAttention achieves nearly the same accuracy as full attention, which demonstrates the effectiveness of our method in complex and dynamic tasks.

It is worth noting that Flat and IVF need to scan 100% and 30% of the past key vectors to achieve the same task accuracy as RetrievalAttention. In contrast, RetrievalAttention only requires scan 1–3% vectors, resulting in much lower decoding latency.

**RULER.** Table 3 demonstrates that models utilizing RetrievalAttention achieve nearly the same task accuracy as full attention in different context lengths. In contrast, other training-free methods experience a noticeable reduction in accuracy, particularly for longer context sizes like 128K, as they fail to capture dynamically changed important tokens.

**Needle-in-a-haystack.** As shown in Figure 5, RetrievalAttention can effectively focus on information at various positions across different context windows, ranging from 4K to 128K. In contrast, other methods like StreamingLLM encounter difficulties when critical information lies beyond the range of the static patterns, whose results are shown in §A.2.

### 4.3 DECODING LATENCY

As the context length increases, the decoding latency of full attention significantly increases due to its quadratic time complexity. Enabling the KV cache (vLLM) incurs out-of-memory (OOM) issues due to limited GPU memory. The latency of StreamingLLM, SnapKV, and InfLLM remains relatively stable because of constant tokens involved in the attention computation, but they suffer significant

Table 3: Performance (%) of different methods and models on RULER.

| | Methods | Act. Tokens | Claimed | Effective | 4K | 8K | 16K | 32K | 64K | 128K | Avg. |
|---|---|---|---|---|---|---|---|---|---|---|---|
| **Llama-3-8B** | FullAttention | 128K | 262K | 32K | 93.13 | 90.49 | 89.27 | 85.11 | 82.51 | 78.74 | 86.54 |
| | StreamingLLM | 2K | - | <4K | 60.10 | 28.77 | 20.99 | 16.36 | 12.52 | 11.34 | 25.01 (-61.53) |
| | SnapKV | 2K | - | 4K | 91.51 | 80.70 | 75.53 | 70.84 | 65.44 | 58.68 | 73.78 (-12.76) |
| | InfLLM | 640+2K | - | 4K | 85.20 | 52.86 | 38.29 | 32.44 | 27.94 | 25.71 | 43.74 (-42.81) |
| | Flat | 640+100 | - | 16K | 92.71 | 87.93 | 87.01 | 84.97 | 80.99 | 74.34 | 84.66 (-1.89) |
| | IVF | 640+100 | - | 16K | 92.73 | 87.86 | 87.22 | 84.74 | 78.46 | 68.21 | 83.20 (-3.34) |
| | **RetrievalAttention** | 640+100 | - | 16K | 92.64 | 88.46 | 86.80 | 84.78 | 80.50 | 74.70 | **84.70(-1.85)** |
| **Yi-9B** | FullAttention | 128K | 200K | 8K | 91.02 | 86.62 | 82.85 | 73.17 | 67.08 | 60.51 | 76.87 |
| | StreamingLLM | 2K | - | <4K | 57.53 | 28.30 | 19.08 | 13.48 | 12.53 | 12.81 | 23.95 (-52.92) |
| | SnapKV | 2K | - | 4K | 90.39 | 75.59 | 64.48 | 48.70 | 39.28 | 32.97 | 58.57 (-18.30) |
| | InfLLM | 640+2K | - | <4K | 82.66 | 50.36 | 36.17 | 28.20 | 22.65 | 20.94 | 40.16 (-36.71) |
| | Flat | 640+100 | - | 8K | 91.09 | 87.71 | 84.42 | 74.58 | 66.16 | 59.50 | 77.24 (+0.37) |
| | IVF | 640+100 | - | 8K | 91.03 | 91.04 | 83.85 | 72.19 | 65.13 | 58.04 | 76.29 (-0.58) |
| | **RetrievalAttention** | 640+100 | - | 8K | 90.78 | 86.32 | 82.95 | 73.73 | 65.67 | 59.15 | **76.43(-0.44)** |
| **Yi-6B** | FullAttention | 128K | 200K | <4K | 84.52 | 77.77 | 69.12 | 61.64 | 58.36 | 55.77 | 67.86 |
| | StreamingLLM | 2K | - | <4K | 51.66 | 24.57 | 15.82 | 9.70 | 9.77 | 11.54 | 20.51 (-47.35) |
| | SnapKV | 2K | - | <4K | 80.94 | 59.55 | 45.36 | 36.11 | 33.43 | 29.53 | 47.49 (-20.37) |
| | InfLLM | 640+2K | - | <4K | 76.42 | 44.38 | 34.11 | 27.11 | 25.28 | 25.33 | 38.77 (-29.09) |
| | Flat | 640+100 | - | <4K | 83.69 | 77.26 | 67.28 | 60.58 | 57.27 | 50.63 | 66.12 (-1.75) |
| | IVF | 640+100 | - | <4K | 83.25 | 76.90 | 67.00 | 58.94 | 55.99 | 50.31 | 63.33 (-2.53) |
| | **RetrievalAttention** | 640+100 | - | <4K | 83.01 | 76.56 | 67.49 | 59.46 | 57.20 | 51.44 | **65.86(-2.00)** |

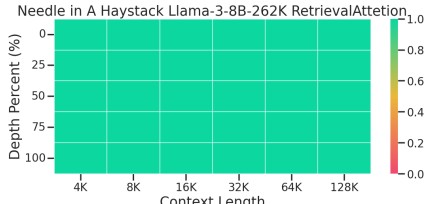

| Methods | 4K | 8K | 16K | 32K | 64K | 128K |
|---|---|---|---|---|---|---|
| Full (without cache) | 0.527 | 1.167 | 2.672 | 6.214 | 15.263 | 43.927 |
| vLLM | OOM | OOM | OOM | OOM | OOM | OOM |
| StreamingLLM | 0.029 | 0.030 | 0.029 | 0.030 | 0.030 | 0.029 |
| SnapKV | 0.029 | 0.028 | 0.028 | 0.029 | 0.029 | 0.028 |
| InfLLM | 0.058 | 0.063 | 0.063 | 0.065 | 0.067 | 0.069 |
| Flat | 0.140 | 0.178 | 0.226 | 0.328 | 0.522 | 0.922 |
| IVF | 0.128 | 0.140 | 0.162 | 0.201 | 0.253 | 0.373 |
| RetrievalAttention | 0.137 | 0.144 | 0.156 | 0.162 | 0.169 | 0.188 |

Figure 5: Performance of RetrievalAttention in Needle-in-a-haystack.

Table 4: Per-token generation latency (s) as context length varies from 4K to 128K on Llama-3-8B.

model accuracy degradation. Due to efficient attention-aware vector search, RetrievalAttention achieves $4.9\times$ and $1.98\times$ latency reduction compared to Flat and IVF for the 128K context.

Table 5 presents the breakdown of end-to-end latency for different retrieval attention-based algorithms under the 128K context length. RetrievalAttention only requires 34.0% of the time for vector search, while Flat and IVF spend 86.6% and 67.0% of time, respectively. This is because RetrievalAttention scans less data for a high recall, avoiding memory bandwidth contention when multiple heads are performing parallel retrieval on the CPU side. Overall, compared with Flat and IVF, RetrievalAttention effectively reduces the index search latency by 91% and 74%, respectively. This advantage becomes more pronounced with longer context lengths.

## 4.4 INDEX RECALL VS. SCANNING VECTORS

Now, we conduct a micro-analysis of the efficiency of attention-aware vector search by examining the relationship between recall and the number of scanned key vectors. The number of key vectors scanned to achieve a target recall serves as an indicator of search efficiency. Figure 6 demonstrates that for the $Q$ to $K$ search, RetrievalAttention requires scanning only a very limited number of key vectors (1–3%) to reach a recall rate higher than 0.95, whereas traditional indexes necessitate retrieving a significantly higher number of keys. We also included a well-known OOD-optimized solution RobustVamana (Jaiswal et al., 2022) for comparison. However, it performs poorly on attention vectors. The efficiency of RetrievalAttention arises because it effectively mitigates the OOD issue between query and key vectors. In contrast, for the in-distribution $K$ to $K$ search, all indexes exhibit good performance.

Table 5: Decoding latency breakdown on Llama-3-8B.

| Methods | Retrieval | Attention | Others | Total |
|---|---|---|---|---|
| Flat | 0.798 | 0.083 | 0.041 | 0.922 |
| IVF | 0.250 | 0.084 | 0.039 | 0.373 |
| RetrievalAttention | 0.064 | 0.081 | 0.043 | 0.188 |

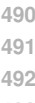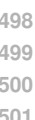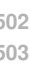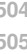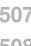

Figure 6: Recall vs. scanning key vectors when using the query vector ($Q$ to $K$) and key vector ($K$ to $K$) as the query, individually. $Q$ and $K$ are dumped from three long-context LLM models.

## 5 RELATED WORKS

To accelerate the long-context LLM inference, some works (Zhang et al., 2023; Liu et al., 2024b; Xiao et al., 2024b; Han et al., 2024; Ge et al., 2024; Li et al., 2024) attempt to compress the size of the KV cache by leveraging the sparsity of attention. However, these methods often suffer from significant model accuracy drops due to the dynamic nature of attention sparsity.

FlexGen (Sheng et al., 2023) and Lamina (Chen et al., 2024b) offload the KV cache to CPU memory, but they struggle with slow and costly full-attention computation. By identifying the dynamic nature of important KV vectors for different queries, recent work chooses to retain all of the KV cache and dynamically attend to different parts of KV vectors based on the current query. Quest (Tang et al., 2024) partitions the KV cache into blocks and selects a representative key vector for each block. For a given query, it scans all representative key vectors and attends top-$k$ blocks with the highest attention scores. InfLLM (Xiao et al., 2024a) adopts a similar strategy as Quest but offloads most KV cache blocks to the CPU memory to support longer contexts. Due to block-based organization and retrieval, the accuracy of representative vectors significantly impacts the effectiveness of those methods for obtaining important tokens. SparQ (Ribar et al., 2024), InfiniGen (Lee et al., 2024), and LoKi (Singhania et al., 2024) approximate the most relevant top-$k$ keys corresponding to a given query by reducing the channel dimension. RetrievalAttention instead organizes the KV cache using ANNS indexes, allowing the retrieval of important tokens with high recalls and low cost. The concurrent work MagicPiG (Chen, 2024) and PQCache (Zhang et al., 2024a) employ LSH and PQ centroids to retrieve critical tokens, respectively. However, they fail to address the OOD issue in attention, necessitating retrieving a large portion of KV cache (e.g., 20%) for high model accuracy.

## 6 CONCLUSION

We propose RetrievalAttention, a method that offloads most KV vectors to CPU memory and leverages vector search for dynamic sparse attention to minimize inference cost. RetrievalAttention identifies the different distributions of the query and key vectors and employs an attention-aware approach to efficiently find critical tokens for model generation. Experimental results demonstrate that RetrievalAttention effectively achieves $4.9\times$ and $1.98\times$ decoding speedup than exact KNN and traditional ANNS methods, on a single RTX4090 GPU for a context of 128K tokens. RetrievalAttention is the first system that supports running 8B-level LLMs with 128K tokens on a single RTX4090 (24GB) GPU with an acceptable latency cost and without compromising model accuracy.

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

# A  ADDITIONAL EXPERIMENTAL DETAILS AND RESULTS

## A.1  MODEL ARCHITECTURE

Table 6 compares the architecture differences of the three models used in our experimental evaluation. All models supports the grouped query attention (GQA), in which multiple query heads share one KV head. Among them, the Yi-9B model has more transformer layers, while the Llama-3-8B model has more KV heads.

Table 6: Architecture overview of LLMs.

| Model | Total Layer | Query Head | KV Head |
|-------|-------------|------------|---------|
| Yi-6B | 32 | 32 | 4 |
| Yi-9B | 48 | 32 | 4 |
| Llama-3-8B | 32 | 32 | 8 |

## A.2  ADDITIONAL RESULTS ON NEEDLE-IN-A-HAYSTACK

Figure 7 shows the results of other methods on Needle-in-a-haystack benchmark. StreamingLLM can only find the correct answer when the needle's position is within the static pattern. InfLLM maintains high performance with shorter context lengths. However, as the length increases, its performance shows a significant decline. Although SnapKV, Flat, and IVF perform well on this benchmark, we have analyzed their disadvantages in accuracy and latency in the previous evaluation.

## A.3  PERFORMANCE IN THE EXTREMELY LONG-CONTEXT INFERENCE

Figure 8 shows the evaluation results of RetrievalAttention for extremely long contexts using the model Llama-3-8B-1048K. RetrievalAttention still passes all test cases when ranging the context length from 250K to 1 million, which demonstrates the robustness of our attention-aware indexes.

## A.4  DECODING LATENCY ON A100

We test the generality of RetrievalAttention by measuring its performance on a server with one A100 (80GB) and one AMD EPYC 7V13 CPU with 24 cores and 220GB DRAM. We show the token-generation latency of different methods on three models in Table 7. Since the KV cache of full attention is disabled, all prompt tokens need to be recalculated during the decoding, incurring a very high decoding latency. By enabling the KV cache with the PageAttention optimization in vLLM, the decoding latency is significantly reduced. However, vLLM suffers from OOM issue with the increase of context length, which we elaborate further later. Other KV cache dropping or block retrieval methods including StreamingLLM, SnapKV, and InfLLM achieve faster decoding speed, but this is at the expense of a significant drop in model accuracy. In contrast, RetrievalAttention does not compromise generation accuracy while achieving much lower decoding latency than IVF and Flat because of the efficient mitigation of out-of-distribution problem.

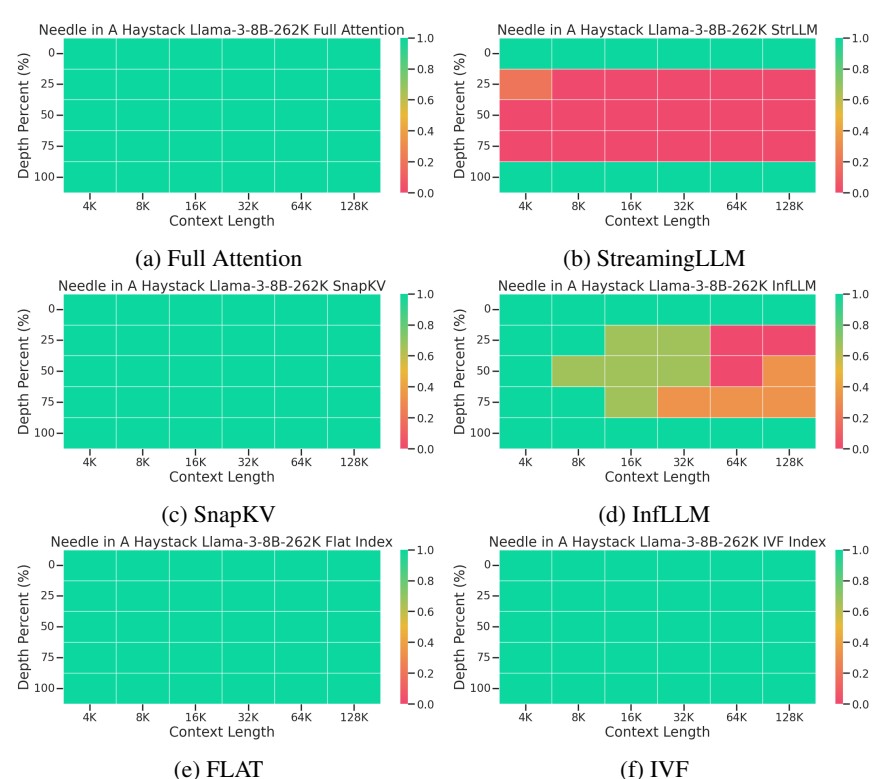

Figure 7: Performance of different algorithms and models on Needle-in-a-haystack. The size of the static pattern is consistently 640 (128 initial tokens + 512 tokens in the local window).

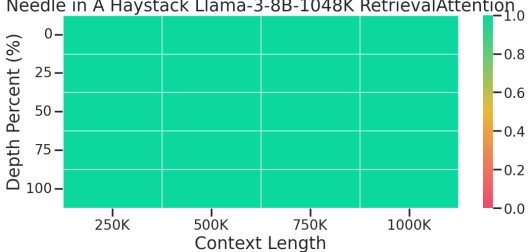

Figure 8: Performance of RetrievalAttention in 1 million Needle-in-a-haystack test.

We also evaluate how the decoding latency changes when the context length varies from 100K to 1M tokens on Llama-3-8B model and the results can be found in Table 8. To make sure there is enough CPU memory to hold the KV cache and indexes, especially in the 1M context scenario, we use a powerful machine equipped with an AMD EPYC 7V12 CPU with 48 cores and 1.72 TB of memory. The machine is also equipped with the same 80GB A100 GPU. The decoding latency of full attention with KV state re-computation increases quadratically with the context size. With the KV cache enabled in the GPU memory, vLLM starts triggering the OOM issues when the context size is larger than 200K. Static KV dropping methods such as StreamingLLM have no latency increase due to the constant KV cache involved for attention computation. Different from Flat and IVF whose latency numbers are sensitive to context size, RetrievalAttention only has a minor latency increase (8%) when the context size increases $10\times$ from 100K to 1M.

Table 7: Per-token generation latency (s) of 128K context-length on A100.

| Methods | Yi-6B | Yi-9B | Llama-3-8B |
|---|---|---|---|
| Full (without cache) | 31.61 | 47.51 | 33.38 |
| vLLM | 0.030 | 0.044 | 0.033 |
| StreamingLLM | 0.032 | 0.047 | 0.031 |
| SnapKV | 0.033 | 0.05 | 0.033 |
| InfLLM | 0.069 | 0.11 | 0.068 |
| Flat | 0.541 | 0.802 | 0.564 |
| IVF | 0.309 | 0.468 | 0.345 |
| RetrievalAttention | 0.150 | 0.227 | 0.155 |

Table 8: Per-token generation latency (s) as context length varies from 100K to 1M.

| Methods | 100K | 200K | 500K | 1M |
|---|---|---|---|---|
| Full (without cache) | 25.47 | 83.03 | 457 | 1740 |
| vLLM | 0.029 | 0.046 | OOM | OOM |
| StreamingLLM | 0.034 | 0.035 | 0.032 | 0.035 |
| SnapKV | 0.035 | 0.035 | 0.034 | 0.034 |
| InfLLM | 0.082 | 0.079 | 0.082 | 0.084 |
| Flat | 0.489 | 0.871 | 1.92 | 3.69 |
| IVF | 0.308 | 0.476 | 1.032 | 1.889 |
| RetrievalAttention | 0.159 | 0.167 | 0.170 | 0.172 |

# B RETRIEVALATTENTION ALGORITHM

## B.1 FORMULA OF COMBINING ATTENTION RESULTS FROM THE CPU AND GPU SIDE

RetrievalAttention partitions the KV vectors for attention into two disjoint sets: predictable ones on GPU (denoted as $\mathcal{W}$) and dynamically retrieved ones on CPU (denoted as $\Omega$).

$$\mathcal{I}_{t,\epsilon} = \mathcal{W} \cup \Omega \tag{3}$$

Attention operation is applied to the two sets of KV vectors separately on CPU and GPU, generating two partial attention outputs (denoted as $\mathbf{o}_{\mathcal{W}}$ and $\mathbf{o}_{\Omega}$, respectively). To guarantee the approximated attention output equals to the attention computation on $\mathcal{I}_{t,\epsilon}$, RetrievalAttention uses a similar idea of FlashAttention (Dao et al., 2022) to combine $\mathbf{o}_{\mathcal{W}}$ and $\mathbf{o}_{\Omega}$ in the following equations:

$$\mathbf{o}_{\mathcal{W}} = \text{Attn}(\mathbf{q}_t, \mathbf{K}[\mathcal{W}, :], \mathbf{V}[\mathcal{W}, :])$$
$$= \frac{\sum_{i \in \mathcal{W}} e^{z_i - \tilde{z}_1} \cdot v_i}{\sum_{i \in \mathcal{W}} e^{z_i - \tilde{z}_1}}$$
$$\mathbf{o}_{\Omega} = \text{Attn}(\mathbf{q}_t, \mathbf{K}[\Omega, :], \mathbf{V}[\Omega, :])$$
$$= \frac{\sum_{i \in \Omega} e^{z_i - \tilde{z}_2} \cdot v_i}{\sum_{i \in \Omega} e^{z_i - \tilde{z}_2}}$$
$$\mathbf{o}_t = \gamma_1 \cdot \mathbf{o}_{\mathcal{W}} + \gamma_2 \cdot \mathbf{o}_{\Omega} \tag{4}$$

where $\tilde{z}_1 = \max_{i \in \mathcal{W}} z_i$ and $\tilde{z}_2 = \max_{i \in \Omega} z_i$ are the local maximum dot products in set $\mathcal{W}$ and $\Omega$ respectively. And $\gamma_1$ and $\gamma_2$ are re-scaling factors to guarantee the attention output is the same as that on $\mathcal{I}_{t,\epsilon}$, which are defined as follows:

$$\gamma_1 = \frac{e^{\tilde{z}_1 - \tilde{z}} \cdot \sum_{i \in \mathcal{W}} e^{z_i - \tilde{z}_1}}{\sum_{i \in \mathcal{I}_{t,\epsilon}} e^{z_i - \tilde{z}}}$$
$$\gamma_2 = \frac{e^{\tilde{z}_2 - \tilde{z}} \cdot \sum_{i \in \Omega} e^{z_i - \tilde{z}_2}}{\sum_{i \in \mathcal{I}_{t,\epsilon}} e^{z_i - \tilde{z}}} \tag{5}$$

---

**Algorithm 1:** RetrievalAttention

---

**Input:** Query vector $\mathbf{q}_t \in \mathcal{R}^{1 \times d}$
**Data:** KV Cache in GPU $\mathbf{K}_\mathcal{W}, \mathbf{V}_\mathcal{W} \in \mathcal{R}^{|\mathcal{W}| \times d}$
**Data:** CPU-based Vector Database $\mathcal{H}$
**Output:** Attention output $\mathbf{o}_t \in \mathcal{R}^{1 \times d}$
```
// Find the predictable KV vectors
```
1 $\mathcal{W}' \leftarrow$ PredictActiveTokens(...);
2 **for** $\{i | i \in \mathcal{H} \cup \mathcal{W}'\}$ **do**
3 $\quad \lfloor \mathcal{H}$.remove(i); $\mathcal{W}$.insert(i); // move to GPU
4 **for** $\{i | i \notin \mathcal{W}' \wedge i \in \mathcal{H}\}$ **do**
5 $\quad \lfloor \mathcal{W}$.remove(i); $\mathcal{H}$.insert(i); // move to CPU
```
// Attention on GPU
```
6 $\mathbf{o}_\mathcal{W} \leftarrow$ FlashAttention($\mathbf{q}_t, \mathbf{K}_\mathcal{W}, \mathbf{V}_\mathcal{W}$)
```
// Attention on CPU
// Search in vector database to retrieve most relevant KV
     vectors
```
7 $\Omega \leftarrow$ VectorSearch($\mathbf{q}_t$);
8 $o_\Omega \leftarrow$ AttentionCPU($\Omega$); // Combine partial attention outputs
9 $\mathbf{o}_t = \gamma_1 \cdot \mathbf{o}_\mathcal{W} + \gamma_2 \cdot \mathbf{o}_\Omega$; // Equation 4,5

---

## B.2 OVERALL EXECUTION FLOW

Algorithm 1 summarizes the above design of RetrievalAttention and elaborates the procedure in an algorithm. At the beginning of each token generation, RetrievalAttention predicts active KV vectors, moves them to GPU memory, and computes partial attention using the FlashAttention (Dao et al., 2022) kernel (#1 - #6). In parallel with GPU computation, RetrievalAttention leverages the specially designed vector database to find the most relevant KV vectors to compute attention on CPU (#7 - #8). Finally, RetrievalAttention combines the partial attention outputs on GPU and CPU using #4 and gets the approximated attention output (#9).

## C IMPLEMENTATION

RetrievalAttention builds one individual vector index for the KV cache in one attention head. RetrievalAttention has implemented several optimizations to optimize the prompt prefill, accelerate the vector search, and reduce CPU memory usage.

**Optimization for the Prefill Phase.** During the prefill phase, full attention computation is required to generate the output vector for the next layer of the LLM. Simultaneously, we move the KV vectors to the CPU side for the ANNS index building. To accelerate the overall prefill process, we overlap the cache movement to the CPU with the full attention computation on the GPU in a pipeline manner. To minimize peak GPU memory usage during the prefill phase, attention computation is performed sequentially across multiple attention heads. This approach only slightly impacts the attention computation speed, as longer prompts can fully leverage GPU parallelism with FlashAttention.

For simplicity, our current implementation does not integrate the KNN computation from Q to K into FlashAttention. However, the KNN computation can be fused into FlashAttention for acceleration because (1) FlashAttention inherently includes the step of calculating the inner product of queries and keys, and (2) the top-K algorithm only consumes registers that are redundant in the flash-attention kernel for modern GPUs. We can initialize a min-heap for each query in registers and update it with the matrix multiplication results as the thread block moves through key blocks. Heap operations are performed in CUDA cores and may take up to 2-3× the prefill latency, which is acceptable. The detailed algorithm is shown in Algorithm **??**.

**Handling Newly Generated Tokens.** In our current implementation, we do not update the index and, consequently, the nearest-keys set for prefilled queries during the decoding phase. Instead, we maintain the newly generated tokens in the GPU memory as a static pattern and include all of them

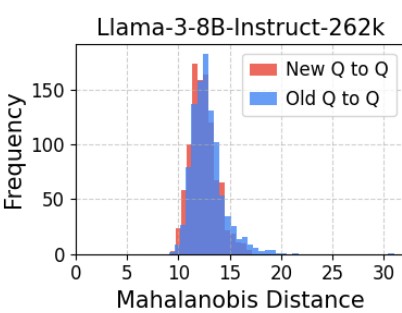

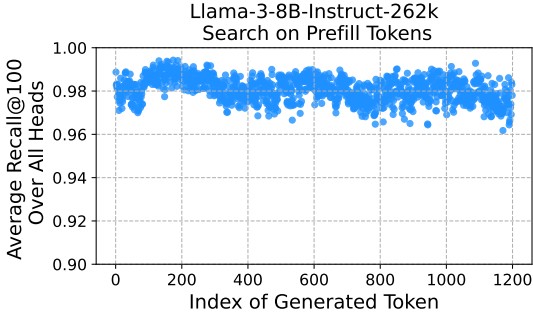

Figure 9: The Mahalanobis distance from the newly generated queries (New Q) and sampled prefill queries (Old Q) to the remaining prefill queries.

Figure 10: Index recalls remain at a high level ($\geq$0.95) in all attention heads when using 1200 newly generated queries to search in the indexes built for prompt context.

in the attention computation for subsequent generation steps. This design choice is based on the observation that newly generated tokens are typically much fewer in number compared to the long context.

The maximum generation length of long-context benchmark Ruler and InfiniBench is 128 tokens. Existing studies (Bai et al., 2024; Zhao et al., 2024) have also demonstrated that although modern LLMs can handle extremely long contexts (e.g., 128K to 1M tokens), the maximum generation length usually does not exceed 2K tokens. Therefore, newly generated tokens can either remain in the GPU memory as static patterns or be offloaded to the CPU memory and efficiently retrieved using a brute-force KNN method. Since we do not update the original index, the index quality for searching critical tokens within the prompt context remains unchanged as the decoding queries follow the same distribution as prefill queries. Figure 9 illustrates that the Mahalanobis distances from the newly generated queries to the prefill queries are nearly identical to the distances among the prefill queries themselves. This confirms that the newly generated queries remain within the distribution of prefill queries. Furthermore, we employ a summary task, which generates 1,200 tokens based on a prompt context of 128K tokens, to validate our design. Figure 10 demonstrates that the index provides robust performance with high recall of top-k results in prefill key vectors when handling new queries in a long generation.

Although we have adopted the above design choices, our ANNS can accept incremental inserts without updating the nearest-keys of prefilled queries. Specifically, we can utilize the update strategy from RoarGraph (Chen et al., 2024a) to achieve this. For each new key vector ($v$), we find the closest prefilled query ($q$), get the previous nearest-key set ($S$), and use it to select neighbors for $v$ without modifying the pre-existing set. This insert strategy is shown to be highly efficient while maintaining the indexing quality (Chen et al., 2024a). To exemplify this, we insert every key vector of the generated token into the index in a summary task and test recall rates among all heads in Llama-3-8B. Figure 11 presents the index of RetrievalAttention maintains its quality at high recall regimes when continuing to insert newly generated key vectors into the index.

To validate the index works well on special cases with short input and relatively long generation, we evaluate the index on code generation tasks from LongBench (Bai et al., 2023). The input lengths for these tasks range from 151 to 719 tokens, with the model allowed to generate up to 2,000 tokens, incorporating incremental inserts into the index. This output length covers most scenarios in existing benchmarks. We evaluate the recall rates of the index at each decoding step. As shown in Figure 12, while recall rates exhibit a slight decline as decoding progresses, the index remains robust, maintaining a recall rate of at least 0.95 in long-generation scenarios, where the output length exceeds the input length by multiple times.

**Multi-head Parallelism on the CPU side.** To speed up the dynamic sparse attention computation on the CPU, we exploit the multi-thread parallelism in vector databases by leveraging the multi-core ability of modern CPU architecture. Specifically, since the computation of different attention heads

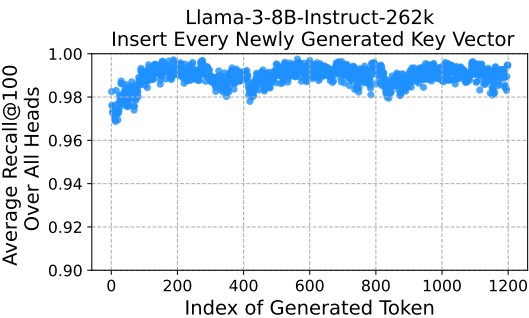

Figure 11: Index recalls remain at a high level among all attention heads when inserting every newly generated key vector into the index in a summary task that generates 1,200 tokens.

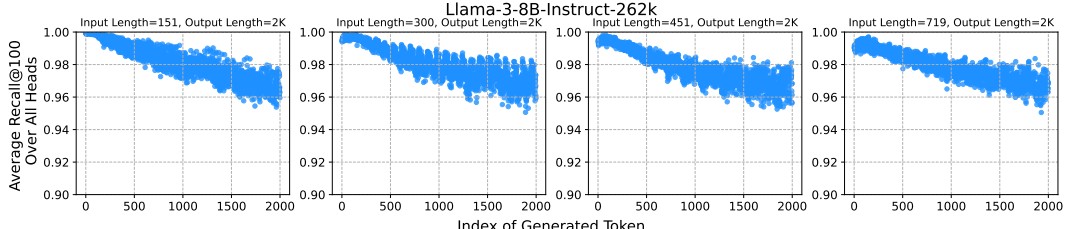

Figure 12: Index recalls on contexts in code generation tasks, with short prompt and long generation.

is independent, we launch multiple threads for parallel searching across different vector indexes to reduce the overall latency on the CPU side. For grouped query attention (GQA) (Ainslie et al., 2023), although multiple query heads could share the same key-value vectors, we observe that the query vectors from different query heads in the same group exhibit different vector distributions. Therefore, we build one vector index for each query head to leverage the specific query distribution of each head.

**Minimize the CPU Memory Usage.**    To reduce CPU memory consumption, the indexes in the same attention group share one copy of KV vectors by only storing the pointers to KV vectors in each index. In the future, we plan to utilize scalar quantization to further compress the KV vectors, implementing an 8-bit quantization in place of the original FP16 format. This compression is promising to reduce memory usage while preserving computational efficiency. Importantly, our initial results demonstrate that this quantization approach does not compromise the inference accuracy, maintaining performance equivalent to the full-precision representation.

## D    ADDITIONAL RELATED WORK

**Sparse Transformers.**    Since the quadratic complexity of attention has become the bottleneck of LLM efficiency for long context applications, numerous works have studied to design sparse transformers to reduce the computational and memory complexity of the self-attention mechanism. Some works restrict the attention computation to predefined patterns, including sliding windows (Child et al., 2019), dilated windows (Beltagy et al., 2020), or a mixture of different patterns (Zaheer et al., 2020; Ainslie et al., 2020). Some approaches use cluster-based sparsity based on hash value (Kitaev et al., 2020) or KNN algorithms (Bertsch et al., 2024; Mao et al., 2024). These solutions either require pre-training a model from scratch or target limited scenarios like CPU-only, which do not work for our target to out-of-box usage of LLMs on the GPU-CPU architecture. Although some approaches (Xiao et al., 2024a; Ribar et al., 2024) exploit the dynamic sparse nature of LLMs, they often use some estimation using low-rank hidden states or post-statistical approaches, which incurs high overhead but with low accuracy. Moreover, all these approaches have to maintain full KV vectors on GPU with only accelerated inference by reduced memory movement, which does not solve the challenge of commodity GPUs with limited GPU memory.

Additionally, some approaches accelerate the inference by employing dynamically sparse attention patterns (Jiang et al., 2024), separating the prefill and decoding stages (Zhong et al., 2024; Qin et al., 2024), and utilizing sequence parallelism (Jacobs et al., 2023; Liu et al., 2024a). These methods are orthogonal to ours and can be in conjunction with our approach.

# E    ADDITIONAL BASELINES

We compare RetrievalAttention with additional baselines InfiniGen and Quest on the RULER benchmark and show the results on Table 9. InfiniGen and Quest exhibit a noticeable drop in model accuracy compared to full attention. In contrast, RetrievalAttention performs best and achieves nearly the same accuracy as full attention across two benchmarks.

Table 9: Performance (%) of different methods in 128K context length on RULER.

| | Methods | Act. Tokens | S1 | S2 | S3 | M1 | M2 | M3 | MQ | MV | VT | CW | FW | Q1 | Q2 | Avg. |
|---|---|---|---|---|---|---|---|---|---|---|---|---|---|---|---|---|
| Llama-3 | FullAttention | 128K | 100.0 | 100.0 | 100.0 | 98.0 | 98.0 | 73.0 | 94.5 | 97.0 | 87.0 | 1.0 | 72.2 | 58.5 | 44.5 | 78.7 |
| | InfiniGen | 2K | 99.0 | 91.5 | 24.5 | 82.5 | 25.0 | 0.0 | 30.3 | 27.8 | 67.3 | 1.2 | 45.5 | 33.0 | 32.5 | 43.1 |
| | Quest | 2K | 100.0 | 100.0 | 98.5 | 98.5 | 36.5 | 0.0 | 48.9 | 64.3 | 89.4 | 1.0 | 64.5 | 45.0 | 39.5 | 60.5 |
| | **Ours** | 640 + 100 | 100.0 | 100.0 | 100.0 | 99.0 | 98.0 | 45.0 | 92.8 | 93.0 | 88.0 | 1.1 | 49.3 | 60.5 | 44.5 | 74.7 |

# F    DYNAMIC RETRIEVAL BUDGET ALLOCATION

We investigated the impact of adjusting the retrieval budget according to the sparsity degree across layers, by adopting the budget allocation policy from PyramidIKV (Cai et al., 2024). Specifically, we compare the performance of the original RetrievalAttention with and without the PyramidKV-based budget allocation strategy on the InfiniteBench benchmark, as shown in RTable 2. Specifically, for the original RetrievalAttention, we set a fixed budget of 2000 tokens for all heads in all layers. In contrast, PyramidKV dynamically adjusts the retrieval size across different layers, allocating more in lower layers and less in higher ones.

The results in Table 10 shows that PyramidKV allocation strategy achieves better performance in Retr.KV tasks, though it slightly decreases performance in the En.QA task. On average, the accuracy slightly surpasses that of the original RetrievalAttention. This indicates that dynamic budget allocation is promising but may require task-specific allocation strategies.

Table 10: Performance (%) of RetrievalAttention and RetrievalAttention w/ PyramidKV in 128K context length.

| Methods | Retr.N | Retr.P | Retr.KV | Code.D | Math.F | En.QA | En.MC | Avg. |
|---|---|---|---|---|---|---|---|---|
| **Full Attention** | 100.0 | 100.0 | 17.5 | 19.0 | 39.5 | 9.1 | 68.0 | 50.4 |
| **RetrievalAttention** | 100.0 | 100.0 | 14.5 | 18.5 | 40.0 | 8.7 | 67.5 | 49.9 |
| **RetrievalAttention w/ PyramidKV** | 100.0 | 100.0 | 16.0 | 18.5 | 40.0 | 8.5 | 67.5 | 50.1 |

# G    PERFORMANCE ON THE LARGER MODEL

To demonstrate the generalizability of our methods on larger models, we evaluated our method on Llama-3-70B-262k using a server with eight 40GB A100 GPUs by partitioning the model by layers across GPUs. We choose the most complex task KV retrieval in ∞-Bench to stress test the efficiency of RetrievalAttention and other baselines.

Table 11: Performance (%) and decoding latency (s) in Llama-3-70B Model.

|                  | Full | StreamingLLM | Quest | Flat | RetrievalAttention |
|------------------|------|--------------|-------|------|--------------------|
| **Accuracy**     | 35.0 | 0.0          | 13.0  | 24.0 | 23.5               |
| **Decoding latency** | 248  | 0.14         | 1.36  | 5.68 | 1.62               |

The results in the Table 11 shows that RetrievalAttention achieves nearly the same task accuracy as the exact KNN method Flat, and outperforms Quest by 80%. The decoding speed of RetrievalAttention is $3.5\times$ faster than Flat as it effectively reduces the vectors to scan.

## H   TopK Kernel Implementation Details

In practical computations, the TopK selection during the Q and K matrix multiplication can be fused directly into the FlashAttention computation, thereby minimizing the overhead of building indices. Specifically, during the FlashAttention operation, BitonicSort (Nassimi & Sahni, 1979) and BitonicMerge (Johnson et al., 2019) algorithms are used within CUDA cores to efficiently retain TopK information, while Tensor Cores are simultaneously utilized for matrix multiplication. This design ensures that, on GPUs such as NVIDIA Hopper, the TopK retrieval process is fully hidden within the FlashAttention computation through the parallel utilization of CUDA cores and Tensor Cores (Shah et al., 2024), resulting in a highly efficient pipeline. Detailed steps are outlined in Algorithm 2.

**Algorithm 2:** Select Top-K key tokens for each query token, fused with Flash-Attention

---

**Shape:** sequence length $S$, head dim $d_h$, top K number $T$, block size $B$, block number $N = \lceil \frac{S}{B} \rceil$

**Input:** $\boldsymbol{Q}, \boldsymbol{K}, \boldsymbol{V} \in \mathbb{R}^{S \times d_h}$

Initialize top K indices $\boldsymbol{J} \leftarrow (0)^{S \times T} \in \mathbb{N}^{S \times T}$

Scale $\tau \leftarrow \sqrt{\frac{1}{d_h}}$

*# Thread-block-level parallelized*
**for** $i \leftarrow 1$ to $N$ **do**
    Load $\boldsymbol{Q}_{\text{chip}} \leftarrow \boldsymbol{Q}^{i \times B:(i+1) \times B} \in \mathbb{R}^{B \times d_h}$ into shared memory

    *# B sorted arries of size T with member format of (value, index)*
    Initialize $\boldsymbol{A} \leftarrow (-\infty, 0)^{B \times T} \in (\mathbb{R}, \mathbb{N})^{B \times T}$ in registers
    Initialize $\boldsymbol{O}_{\text{chip}} \leftarrow (0)^{B \times d_h} \in \mathbb{R}^{B \times d_h}$ in shared memory
    Initialize $\boldsymbol{m} \leftarrow (-\inf)^B \in \mathbb{R}^B$ in registers
    Initialize $\boldsymbol{l} \leftarrow (0)^B \in \mathbb{R}^B$ in registers

    *# Loop through K, causal*
    **for** $j \leftarrow 1$ to $i$ **do**
        Load $\boldsymbol{K}_{\text{chip}} \leftarrow \boldsymbol{K}^{j \times B:(j+1) \times B} \in \mathbb{R}^{B \times d_h}$ into shared memory
        Load $\boldsymbol{V}_{\text{chip}} \leftarrow \boldsymbol{V}^{j \times B:(j+1) \times B} \in \mathbb{R}^{B \times d_h}$ into shared memory

        *# Calculate $QK^T$ in tensor cores*
        $\boldsymbol{S} \leftarrow \tau \boldsymbol{Q}_{\text{chip}} \boldsymbol{K}_{\text{chip}}^{\text{T}}$
        $\boldsymbol{S} \leftarrow \text{mask}(\boldsymbol{S})$

        *# Top-K in CUDA cores*
        $\boldsymbol{A}_{new}^{ii} \leftarrow \text{BitonicSort}(\boldsymbol{S}^{ii})$
        $\boldsymbol{A}^{ii} \leftarrow \text{BitonicMerge}(\boldsymbol{A}^{ii}, \boldsymbol{A}_{new}^{ii})$

        *# Online softmax in CUDA cores*
        $\boldsymbol{m}_{new}^i \leftarrow \max(\boldsymbol{m}^i, \text{rowmax}(\boldsymbol{S})) \in \mathbb{R}^B$
        $\boldsymbol{S} \leftarrow \boldsymbol{S} - \boldsymbol{m}_{new}^i$
        $\boldsymbol{P} \leftarrow \exp(\boldsymbol{S})$
        $\boldsymbol{l}_{new}^i \leftarrow \text{rowsum}(\boldsymbol{S})$
        $\boldsymbol{\alpha} \leftarrow \exp(\boldsymbol{m}^i - \boldsymbol{m}_{new}^i)$
        $\boldsymbol{l}^i \leftarrow \boldsymbol{\alpha} \boldsymbol{l}^i + \boldsymbol{l}_{new}^i$

        *# Calculate PV in tensor cores*
        $\boldsymbol{O}_{\text{chip}} \leftarrow \boldsymbol{\alpha} \boldsymbol{O}_{\text{chip}} + \boldsymbol{P} \boldsymbol{V}_{\text{chip}}$
    **end for**

    *# Write top-K outputs, thread-level parallelized*
    **for** $ii \leftarrow 1$ to $B$ **do**
        Save $\boldsymbol{J}^{i \times B + ii} \leftarrow \boldsymbol{H}^{ii}.\text{indices}$
    **end for**

    *# Write flash-attention outputs*
    $\boldsymbol{O}_{\text{chip}} \leftarrow \text{diag}(\boldsymbol{l}^i)^{-1} \boldsymbol{O}_{\text{chip}}$
    Save $\boldsymbol{O}^{i \times B:(i+1) \times B} \leftarrow \boldsymbol{O}_{\text{chip}}$
**end for**

---

