# OpenReview forum: "RetrievalAttention: Accelerating Long-Context LLM Inference via Vector Retrieval"
_ICLR.cc/2025/Conference — Submitted to ICLR 2025_

### Official Review · Reviewer_Bxgi · 2024-10-20

**Soundness:** 2
**Presentation:** 2
**Contribution:** 2
**Rating:** 3
**Confidence:** 5

**Summary:**

This paper introduces a new KV-cache optimization method called RetrievalAttention, which accelerates attention computation and reduces GPU memory consumption. It leverages the dynamic sparsity of the attention mechanism. It uses an attention-aware vector search algorithm that can adapt to the distribution of query vectors, to build indexes for KV vectors in CPU memory and retrieve the most important ones through search during the token generation. The authors compare the performance of RetrievalAttention in long-context LLM inference against full attention and other state-of-the-art methods. The comparisons mainly focus on the accuracy and latency of the proposed approach.

**Strengths:**

1. The task this paper tries to solve is crucial. A longer context window for LLMs allows the model to (1) enhance the consistency and coherence flow of information across a larger span of text; (2) improve understanding of complex text; (3) handle multi-step reasoning.
2. The idea of CPU-GPU offloading for LLM KV-cache optimization is interesting. Although there are several works already contributing in similar directions, they mainly focus on different aspects such as channel-dimension reduction and block-based organization.
3. Identify the challenge of OOD in using the ANNS index for attention computation.

**Weaknesses:**

1. Some details in the method section are unclear and could benefit from further elaboration, particularly regarding the ANNS algorithm and CPU-GPU co-execution. Please refer to Question (1) for more specific inquiries.
2. Certain baselines are missing, such as QUEST[1].
3. I couldn't find the code implementation of the proposed method, which makes it difficult to assess its performance in practice.

**Questions:**

1. Will the nearest-keys set for prefilled queries (indexed during the prefill phase) be updated during token generation? If not, will the retrieval recall degrade as more new tokens are generated?
2. Even if the nearest-keys set is updated, in extreme cases like code generation, where there are very few prefilled tokens and many newly generated tokens, how can the method ensure that the prefilled tokens remain representative enough for the query vectors of the newly generated tokens to effectively use them as a bridge to find the nearest key vectors?
3. For CPU-GPU co-execution, it is unclear how to separate the two disjoint sets of KV cache vectors between CPU and GPU. From my understanding, it seems challenging to ensure that the significant keys selected by ANNS are not included in the set of fixed initial tokens or within the last sliding window.
4. Although does not support GQA, I believe it is still worthwhile to include QUEST[1] as one of the baselines for comparison with vanilla attention.

I am willing to raise my score if the authors can address the limitations mentioned above.

References:
[1] https://github.com/mit-han-lab/Quest/tree/main

---

> ### Author Response · Authors · 2024-11-21
> **Official Comment by Authors (1/2)**
>
> ***W1.** "Some details in the method section are unclear and could benefit from further elaboration, particularly regarding the ANNS algorithm and CPU-GPU co-execution. Please refer to Question (1) for more specific inquiries."*
>
> ***Q1.** "Will the nearest-keys set for prefilled queries (indexed during the prefill phase) be updated during token generation? If not, will the retrieval recall degrade as more new tokens are generated?"*
>
> **Response:** Thank you for your detailed comments. In our current implementation, we do not update the index and, consequently, the nearest-keys set for prefilled queries during the decoding phase. Instead, we maintain the newly generated tokens in the GPU memory as a static pattern and include all of them in the attention computation for subsequent generation steps. This design choice is based on the observation that newly generated tokens are typically much fewer in number compared to the long context.
>
> The maximum generation length of long-context benchmark Ruler and InfiniteBench is 128 tokens. Existing studies [13][14] have also demonstrated that although modern LLMs can handle extremely long contexts (e.g., 128K to 1M tokens), the maximum generation length usually does not exceed 2K tokens. Therefore, newly generated tokens can either remain in the GPU memory as static patterns or be offloaded to the CPU memory and efficiently retrieved using a brute-force KNN method. Since we do not update the original index, the index quality for searching critical tokens within the prompt context remains unchanged as the decoding queries follow the same distribution as prefilled queries. To confirm this, we add Figure 9 in our revised paper, which illustrates that the Mahalanobis distances from the newly generated queries to the prefill queries are nearly identical to the distances among the prefill queries themselves.
>
> Although we have adopted above design choices, our ANNS can accept incremental inserts without updating the nearest-keys of prefilled queries. Specifically, we can utilize the update strategy from RoarGraph [15] to achieve this. For each new key vector ($v$), we find the closest prefilled query ($q$), get the previous nearest-key set ($S$), and use it to select neighbors for $v$ without modifying the pre-existing set. This insert strategy is shown to be highly efficient while maintaining the indexing quality [15].
>
> We have clarified these details in the revised paper.
>
> ***Q2.** "Even if the nearest-keys set is updated, in extreme cases like code generation, where there are very few prefilled tokens and many newly generated tokens, how can the method ensure that the prefilled tokens remain representative enough for the query vectors of the newly generated tokens to effectively use them as a bridge to find the nearest key vectors?"*
>
> **Response:** Thank you for your comment. As stated in the introduction of our paper, we focus on scenarios involving long contexts, where the prefilled queries should be sufficiently large to represent the distribution of queries. In cases like the one you mentioned, since the generation limit of all LLMs is typically small (<2K tokens), it is feasible to perform full attention over the newly generated tokens or offload them to the CPU memory for retrieval using a brute-force KNN method, which remains efficient for a small number of tokens.
>
> [13] WildChat: 1M ChatGPT Interaction Logs in the Wild, ICLR 2024
>
> [14] LongWriter: Unleashing 10,000+ Word Generation From Long Context LLMs, 2024
>
> [15] RoarGraph: A Projected Bipartite Graph for Efficient Cross-Modal Approximate Nearest Neighbor Search, VLDB 2024

---

> ### Author Response · Authors · 2024-11-21
> **Official Comment by Authors (2/2)**
>
> ***W2.** "Certain baselines are missing, such as QUEST."*
>
> ***Q4.** "Although does not support GQA, I believe it is still worthwhile to include QUEST as one of the baselines for comparison with vanilla attention."*
>
> **Response:** Thanks for your suggestion. We have included Quest as a new baseline. We have also added another baseline InfiniGen, as requested by another reviewer. We evaluated their performance on InfiniteBench and RULER, with the results presented in RTable 3 and RTable 4, respectively.
>
> InfiniGen exhibits a noticeable drop in model accuracy compared to full attention due to inaccurate speculation of important tokens from previous layers. Although Quest performs better than InfiniGen, it achieves zero accuracy in the highly dynamic KV retrieval task (Retr.KV in the InfiniteBench) due to low representativeness of centroid vectors. Moreover, both Quest and InfiniGen have severe performance degradation on the RULER benchmark. In contrast, RetrievalAttention performs best and achieves nearly the same accuracy as full attention across two benchmarks.
>
> Regarding the decoding latency, Quest encounters OOM issues on the RTX 4090 due to its fully GPU-resident design, making it unsuitable for supporting 128K contexts in consumer-grade GPUs. Since the current decoding optimizations of InfiniGen does not support GQA for LLMs [8], we will implement InfiniGen for the GQA model and test its decoding latency in the future version of our paper.
>
> |Methods|Act. Tokens|Retr.N|Retr.P|Retr.KV|Code.D|Math.F|En.QA|En.MC|Avg.|
> |-|-|-|-|-|-|-|-|-|-|
> |FullAttention|128K|100.0|100.0|17.5|19.0|39.5|9.1|68.0|50.4|
> |InfiniGen|2K|99.5|100.0|0|17.5|39.0|7.3|57.5|45.8|
> |Quest|2K|100.0|100.0|0|18.0|40.0|8.2|67.0|47.6|
> |Ours|640+100/2K|100.0|100.0|9.0/14.0|19.0|40.0|7.5|67.0|48.9/49.6|
>
> RTable 3. Performance (%) of different methods in 128K context length on InfiniteBench using the Llama-3-8B-262K.
>
>
> |Methods|Act. Tokens|S1|S2|S3|M1|M2|M3|MQ|MV|VT|CW|FW|Q1|Q2|Avg.|
> |-|-|-|-|-|-|-|-|-|-|-|-|-|-|-|-|
> |FullAttention|128K|100.0|100.0|100.0|98.0|98.0|73.0|94.5|97.0|87.0|1.0|72.2|58.5|44.5|78.7|
> |InfiniGen|2K|99.0|91.5|24.5|82.5|25.0|0.0|30.3|27.8|67.3|1.2|45.4|33.0|32.5|43.1|
> |Quest|2K|100.0|100.0|98.5|98.5|36.5|0.0|48.9|64.3|89.4|1.0|64.5|45.0|39.5|60.5|
> |Ours|640+100|100.0|100.0|100.0|99.0|98.0|45.0|92.8|93.0|88.0|1.1|49.3|60.5|44.5|74.7|
>
> RTable 4. Performance (%) of different methods in 128K context length on RULER using the Llama-3-8B-262K.
>
> **W3.** *"I couldn't find the code implementation of the proposed method, which makes it difficult to assess its performance in practice."*
>
> **Response:** We are in the legal process to make our code public. Will release once it is ready.
>
> ***Q3.**"For CPU-GPU co-execution, it is unclear how to separate the two disjoint sets of KV cache vectors between CPU and GPU..."*
>
> **Response:** In the design of RetrievalAttention, during the prefilling phase, we **physically separate** consistently important tokens and dynamic tokens by placing them in GPU memory and CPU memory, respectively. The ANNS index is built only for the remaining dynamic tokens in the CPU memory. Therefore, there is no overlap between static tokens in the GPU cache and dynamic critical tokens retrieved from ANNS indexes. The accurate identification of consistently important tokens is well-studied and is orthogonal to the design of RetrievalAttention. Currently, we adopt StreamingLLM to keep initial and last window tokens in the GPU cache, but we are adaptable to support more complex static patterns. We'll include this in the next version of the paper.

---

> ### Author Response · Authors · 2024-11-25
> **Follow up**
>
> Thank you again for reviewing our work! Please let us know if we have sufficiently addressed your questions and concerns.
>
> Authors

---

> > ### Comment · Reviewer_Bxgi · 2024-11-25
> >
> > Thank you to the authors for their detailed response and the effort invested in conducting additional experiments. I am satisfied with the replies to Q3 and Q4. However, regarding Q1 and Q2, I still have some concerns. Specifically, as noted in the authors’ response:
> > > "it is feasible to perform full attention over the newly generated tokens or offload them to the CPU memory for retrieval using a brute-force KNN method, which remains efficient for a small number of tokens."
> >
> > I think neither performing full attention nor using a brute-force KNN method on the CPU is an acceptable solution, particularly when the number of generated tokens increases significantly.
> >
> > Regarding W3 (code implementation), I will leave it on hold for now.
> >
> > In summary, I am partially satisfied with the authors' reply and will maintain my current score.

---

> ### Author Response · Authors · 2024-11-26
>
> Thank you for your valuable feedback. We apologize for any confusion and would like to address your concerns as follows:
>
> - **Clarification of Attention Computation**: We realize the description of _"full attention over the newly generated tokens"_ in our previous reply was unclear and may have led to misunderstandings. Our current implementation incorporates newly generated tokens into the local window of the static pattern. However, the attention computation over the prompt context always relies on ANNS for dynamic sparse attention.
>
> - **Design Choice Rationale**: The design choice mentioned above was made with a focus on long-context scenarios where the generation lengths are generally small (fewer than 2K tokens), compared to input tokens (e.g., 128K). This is a common setting in existing long-context benchmarks and real-world scenarios, like repo-level code debuging, long-document QA. The strong performance on downstream tasks, as demonstrated in our paper, supports the effectiveness of our method for these scenarios. Moreover, as the generation length increases, the index quality does not decrease even without updating the index. We demonstrate this by adding Figure 10 in our revised paper, where we generate 1200 tokens from the 128K context using a summary task. Figure 10 shows the index recalls on prefill tokens consistently remain at a high level (>0.95) throughout the decoding steps from the first token to the 1200th token. This results is corresponding to Figure 9, which we added previously.
>
> - **Ability to Handle Long Generation**: As the number of generated tokens increases, we can insert the newly generated tokens into the index by adopting the update strategy from RoarGraph [15] instead of keeping them in the GPU cache. To show this, we have implemented the insert operation and added Figure 11 in our revised paper. In this experiment, we proactively insert the key vector of every newly generated token into the index at each decoding step, and recall rates were evaluated among all existing tokens. The variation of inserted index recall at each decoding step can be observed in Figure 11, the index recalls remain high (>0.95) across the whole 1200 decoding steps with a slight increase (6%) of per-token generation latency. This demonstrates that handling newly generated tokens is **NOT** a fundamental limitation of our approach.
>
> - **Future Work on Extremely Long Generation**: We acknowledge that future LLMs may extend their generation lengths beyond 2K, particularly in models like O1 utilizing self-play reasoning. Therefore, we consider the study of extremely long generation as an important area for future work.
>
> - **Primary Contribution**: Finally, we emphasize the primary contribution of our work: **introducing vector retrieval methods into attention computation for long-context scenarios** (Reviewer #MYWT, #Afco, #boJw). We designed a CPU-GPU framework utilizing OOD-aware vector indexes to address distribution discrepancies between queries and keys. While our current design choices impose limitations for certain long-generation scenarios, this is not the primary focus of our paper. These limitations can be addressed through incremental inserts to the index, as supported by our experiments (Figure 11 in our revised paper).
>
> We hope that above points are helpful to clarify your concerns.

---

> ### Comment · Reviewer_Bxgi · 2024-11-27
>
> Thank you to the authors for their detailed response. I would like to kindly reiterate my main concern: when the initial prompt length is short and the expected token generation length is long (such as code generation and multi-round dialogue), the proposed approach may encounter issues due to the two concerns outlined in my original review:
> 1. The nearest-keys set for prefilled queries (indexed during the prefill phase) is not updated during token generation.
> 2. The prefilled tokens may not remain representative enough for the query vectors of the newly generated tokens.
>
> Unfortunately, the newest response from the authors does not fully address these two points.
>
> Specifically, while the authors note that:
>
> > the attention computation over the prompt context always relies on ANNS for dynamic sparse attention.
>
> This explanation still leaves my original concerns unresolved. Additionally, I would like to clarify that I disagreed that *"full attention over the newly generated tokens"* in the authors' first round comment is unclear.
>
> Regarding the example provided:
>
> > Where we generate 1200 tokens from the 128K context using a summary task.
>
> I believe a context length of 128K is too long in light of my concerns.
>
> On the third point regarding the improved method (insert newly generated tokens into the index), I appreciate the authors’ solution, as it is a promising approach to address my concern. However, could the authors also provide details about the latency associated with this solution?
>
> Please feel free to point any possible error in my response out.

---

> ### Author Response · Authors · 2024-11-27
>
> Thanks for your reply. While the scenario of very short input lengths with long outputs is not the primary focus of our paper, we have nonetheless conducted additional experiments, as suggested by the reviewer. It is important to note that our previous demonstrations and the revised paper have already established the index's capability to handle long-generation tasks within a 128K long-context. We further address the reviewer's concern in the following two aspects.
>
> - **Insertion latency.** By conducting incremental inserts, it takes a 6% latency increase in per token generation, making 0.188s per token to 0.199s per token.
>
> - **Short Input Length and Long Generation.** To assess the performance of the index on tasks involving short inputs and relatively long generations, we conducted experiments using the **code generation** task from LongBench. Specifically, we varied the input lengths between 151 and 719 tokens and allowed the model to generate up to 2,000 tokens. We conducted the incremental inserts to the index and measured the index recall rates at each step of the decoding process. As shown in the newly added Figure 12 in our revised paper, although there is a slight decline in recall rates as the generation progresses, the index maintains a high recall rate (> 0.95) even in long-generation scenarios where **the output length significantly exceeds the input by multiple times ($2.8\times$ to $13\times$)**. This underscores the index's ability to effectively handle long generation tasks while preserving high recall, even as output length significantly grows.
>
> Regarding two core concerns from the reviewer, (1) The nearest-keys set for prefilled queries is NOT updated during token generation, instead, we choose the lightweight update strategy from RoarGraph, (2) The above experiment demonstrates that the prefilled queries are representative enough for newly generated queries, even in the case that prefill context is very short.

---

> > ### Comment · Reviewer_Bxgi · 2024-11-28
> >
> > I appreciate the authors for responding to my concerns and conducting additional experiments.
> >
> > While I agree with the stated motivation of the paper, some additional revisions are required before it's ready for publication at a venue like ICLR:
> >
> > 1. most of the original experiments are on long-prefill settings only.
> > 2. the original method has the potential risk of performance degradation during long-generation scenarios.
> > 3. rerunning (at least) some of the experiments with the modified version (the one with incremental insertion), and reporting both latency and accuracy results.
> > 4. providing more details of the method or releasing the codebase for evaluation.
> >
> > I believe these revisions are essential to demonstrate that the method is both general and applicable across a range of task settings, while achieving performance comparable to or exceeding existing methods in this domain. As a result, I maintain my original rating for this work.

---

> ### Author Response · Authors · 2024-11-29
>
> Thank you for your detailed review. However, we respectfully disagree with the characterization of long-generation scenarios as a weakness of our work. Below, we clarify our reasons:
>
> 1. **"Out of Scope"**:
> We want to reiterate that our work focuses on long-context scenarios (i.e., long inputs with short outputs). This is consistent with the scope of **related works[1-4]** and **contemporary studies[5-7]** in the field. For example, SnapKV mentions in Sec. 1[3]: _"In practical applications like chatbots and agents, ..., prompts are often much **larger** than generated responses..."_ While long-generation scenarios are undoubtedly important, they remain largely **unexplored** in the current community, with no established benchmarks, designs, or experiments addressing this issue.
>
> 2. **"Feasible Solution Proposal"**:
> We appreciate the reviewer’s suggestion. During the rebuttal period, we supplemented our work with a proposed solution for long-generation scenarios, utilizing an incremental insertion strategy. We included an analysis of attention recall for varying input sizes when generating up to 2k tokens. As suggested by the reviewer, now we have rerun the experiments using incremental inserts on KV retrieval, which is one of the most challenging task on long context, to further validate the approach shown in **RTable 5**. Due to time limit, we will include more experiments in future versions of the paper.
>
> |Methods|Retr.KV|Decoding Latency (s/token)|
> |-|-|-|
> |FullAttention|17.5|43.927|
> |Quest|0|-|
> |RetrievalAttention|14.0|0.188|
> |RetrievalAttention w/ insertion |14.0|0.199|
>
> **RTable 5.** _Performance on KV retrieval task in InfiniteBench using Llama-3-8B-262K in RTX 4090. We cannot get the decoding latency of Quest because it incurs OOM issue in RTX 4090._
>
> Additionally, our paper comprehensively demonstrates the rationale of our insights and the effectiveness of our proposed method. Compared to baselines (e.g., Quest, InfiniGen), our approach achieves higher accuracy and lower decoding latency across various datasets and long-context LLMs.
>
> Regarding the code release, we are committed to releasing it promptly after the review process. It is important to note that releasing the code during the submission phase is not a mandatory requirement for ICLR, and therefore, we do not consider this a weakness of our paper. Additionally, our paper provides a comprehensive explanation of the implementation and experimental setup in Sections 3.3, 4.1, and Appendix C, ensuring the reproducibility of our results.
>
> Finally, we trust that the reviewer will fairly reassess the paper and adjust the score upon addressing your concerns.
>
>
> [1] H2O: Heavy-Hitter Oracle for Efficient Generative Inference of Large Language Models, NeurIPS 2023.
> [2] Efficient Streaming Language Models with Attention Sinks, ICLR 2024.
> [3] SnapKV: LLM Knows What You are Looking for Before Generation, NeurIPS 2024.
> [4] Quest: Query-Aware Sparsity for Efficient Long-Context LLM Inference, ICML 2024.
> [5] MagicPIG: LSH Sampling for Efficient LLM Generation, 2024.
> [6] PQCache: Product Quantization-based KVCache for Long Context LLM Inference, 2024.
> [7] Squeezed Attention: Accelerating Long Context Length LLM Inference, 2024.

---

> > ### Author Response · Authors · 2024-12-02
> > **Kind Request for Prompt Feedback as Discussion Period Concludes**
> >
> > We are deeply grateful for your reviews on our paper. As the discussion period is closing soon, we would sincerely appreciate your thoughts on our recent response. We have extensively conducted the experiments as per your suggestions and kindly request that you consider adjusting your scores accordingly.
> >
> > Your evaluation is vital for the success of our paper. Thank you very much for your time and consideration.
> >
> > Sincerely,
> > Authors

---

> ### Author Response · Authors · 2024-12-03
>
> Due to the approaching rebuttal deadline, we want to ensure that we have addressed your concerns one by one.
>
> **In the First Round of Rebuttal**
>
> The reviewer is **_satisfied_** with our additional experiments:
>
> 1. Successfully clarifying CPU-GPU co-execution
> 2. Comparing with Quest (accuracy: 60.5%, 2K tokens) and InfiniGen (acquired by R3, accuracy: 43.1%, 2K tokens). In contrast, we only activate 100 + 640 tokens, achieving 74.7% accuracy, which is the closest to full attention on RULER benchmark (RTable 4, Table 9).
>
> **In the Subsequent Comments**
>
> **_First, the reviewer is concerned about the robustness of our index when facing long-generation tasks._**
>
> We provided experiments with 128K inputs and 1.2K outputs (where 1.2K represents a specific long-generation case in modern benchmarks) on standard datasets. These experiments show that the data distribution during the prefill and decoding phases is similar (Figure 9), and our index is robust to *insertions* (Figures 10 and 11). These results suggest that our index performs effectively even in long-generation tasks.
>
> **_Second, The reviewer is concerned about the input context being too long and requests a scenario with very short input and long output._**
>
> Although we cited relevant works showing that such cases are **rarely evaluated**, and **no well-established benchmarks** exist for them, we conducted additional experiments on a code generation task from LongBench.
> In this task, the model generates outputs of up to 2K tokens from inputs ranging from 155 to 719 tokens. This scenario, where the output length is significantly longer than the input (ranging from **2.8X to 13X**), aligns closely with the reviewer’s "extremely short input and long-generation" concern. We inserted all newly generated key vector into our indexes at each decoding step and evaluate the index recall. As shown in *Figure 12* of our revised paper, our index maintains a high recall rate (> 0.95) even in the long-generation scenarios.
>
> **_Subsequently, the reviewer expressed concern that most of our experiments in original submissiopn "only" focus on long-context._**
>
> We presented detailed evaluation results for context lengths ranging from 4K to 128K in the **initial version** of our paper (Table 3, Table 4, Figure 7), which covers a broad range of contexts *used by most of related works*. Besides, the title of the paper clearly indicates that the focus is on long-context scenarios, which are the primary objective of our study.
>
> After providing additional experiments, **_the reviewer still raised concerns about the "potential risk" of performance in the reviewer's constrainted secenarios -- extremely short input and long-generation_**.
>
> Despite of presenting additional evaluations, we clearly stated that it is an "out-of-scope" and subjective opinion since it is inconsistent with all related works.
>
> And then, **_the reviewer asked to rerun experiment with insertion_** for downstream tasks.
>
> According to our design and additional experiments (Figures 9, 10, 11), the task accuracy of RetrievalAttention remains robust. In response to the reviewer’s request, we reran a challenging long-context task, KV retrieval, involving insertions and observed no accuracy drop in this scenario (see RTable 5), with slight increased latency.

---

### Official Review · Reviewer_boJw · 2024-10-31

**Soundness:** 2
**Presentation:** 3
**Contribution:** 2
**Rating:** 3
**Confidence:** 5

**Summary:**

The authors propose a method that offloads KV Cache to the CPU combined with an ANNS index for sparse attention computation, aim for addressing GPU memory burdens during inference and accelerating decoding speed.

**Strengths:**

1. The approach of using Approximate Vector Search is intuitive and appears reasonable in this offloading attention computation.
2. The authors propose a simple but effective ANNS index that addresses potential OOD issues in this scenario.
3. This method outperforms baseline methods in multiple benchmarks.

**Weaknesses:**

1. It seems that this work relies heavily on obtaining global attention weights during the prefilling process to build the ANNS index. This may be incompatible with the existing FlashAttention technique, which is essential for efficient long-sequence prefilling computation. FlashAttention reduces prefilling overhead by using computation fusion, which avoids explicitly computing global attention weights, yet provides final outputs directly—making it a commonly used approach in long-sequence inference. Empirically, this issue will significantly increase the computing time and memory usage in the prefilling phase. In evaluation, the authors only provided decoding latency and did not consider potential additional overhead caused by the incompatibility with FlashAttention. Since other baselines generally do not encounter this issue, the authors should include experimental results to address these concerns.
2. The article should adopt a more appropriate baseline. Current baselines, such as SnapKV and StreamingLLM, discard a substantial portion of the KV Cache compared to the fully stored KV Cache method used in this paper, predictably leading to lower accuracy. For this work, a more suitable baseline for comparison with the fully stored Cache and offloading method would be:
(1.) Quest[1]: Quest also fully stores the KV Cache and retrieves several important cache entries during decoding. Although the authors note that Quest’s implementation does not support GQA and thus was excluded, since SnapKV can be integrated with GQA, the authors should consider implementing Quest with GQA support or explain why such an implementation would be challenging. (2.) InfiniGen[2]: InfiniGen is a comparable approach that offloads the Cache to the CPU, retrieving only a subset of the Cache during decoding to achieve approximate decoding, resulting in GPU memory savings and accelerated performance. This would likely be the closest baseline to the method in this paper.
3. It appears that this paper does not account for the overhead of index construction, which also directly impacts overall inference time. The authors should provide results showing the time and memory required for index construction in different input lengths.

[1] Tang, Jiaming, et al. "Quest: Query-Aware Sparsity for Efficient Long-Context LLM Inference." arXiv preprint arXiv:2406.10774 (2024).

[2] Lee, Wonbeom, et al. "{InfiniGen}: Efficient generative inference of large language models with dynamic {KV} cache management." 18th USENIX Symposium on Operating Systems Design and Implementation (OSDI 24). 2024.

**Questions:**

Does sequence length during prefilling impact the effectiveness of the index? Intuitively, longer sequences would result in larger indexes, suggesting that variations in sequence length could affect index performance. Testing the index's effectiveness in needle in a Haystack by locating the 'needle' cache across different sequence lengths could help clarify this effect.

---

> ### Author Response · Authors · 2024-11-21
> **Official Comment by Authors (1/2)**
>
> Thanks for your detailed and insightful comments, which have been very helpful for our work.
>
> ***W1.** "It seems that this work relies heavily on obtaining global attention weights during the prefilling process to build the ANNS index. This may be incompatible with the existing FlashAttention technique..."*
>
> **Response:** As stated in the beginning of Section 3, this paper focuses on the decoding phase by assuming that prefill and index construction are performed in advance (please refer to the response to W3 below for more details). Therefore, for simplicity, our current implementation does not integrate the KNN computation from Q to K into FlashAttention. Instead, we are using a brute-force method that relies on Faiss Flat [7] for KNN computation on the GPU.
>
> However, the KNN computation can be fused into FlashAttention because: (1) FlashAttention inherently includes the step of calculating the inner product of querys and keys, and (2) the top-*K* algorithm only consumes registers that are redundant in flash-attention kernel for modern GPUs. We can initialize a min-heap for each query in registers and update it with the matrix multiplication results as the thread block moves through key blocks. Heap operations are performed in CUDA cores and may take up to 2-3$\times$ the prefill latency, which is acceptable. More details are provided in the Algorithm 2 in our revised paper. Exisitng KV cache compression work such as H2O [8] also requires KNN computation during the prefilling phase. Therefore we regard optimizing the KNN computation as important future work to benefit broader research works.
>
> ***W2.** "The article should adopt a more appropriate baseline...Quest...InfiniGen..."*
>
> **Response:** Thanks for your suggestion. In the revised paper, we have added Quest and InfiniGen as new baselines. We evaluated their performance on InfiniteBench and RULER, with the results presented in RTable 3 and RTable 4, respectively.
>
> InfiniGen exhibits a noticeable drop in model accuracy compared to full attention due to inaccurate speculation of important tokens from previous layers. Although Quest shows better performance than InfiniGen, it achieves zero accuracy in the highly dynamic KV retrieval task (Retr.KV in the InfiniteBench) due to low representativeness of centroid vectors. Moreover, both Quest and InfiniGen have severe performance degradation on the RULER benchmark. In contrast, RetrievalAttention performs best and achieves nearly the same accuracy as full attention across two benchmarks.
>
> Regarding the decoding latency, Quest encounters OOM issues on the RTX 4090 due to its fully GPU-resident design, making it unsuitable for supporting 128K contexts in consumer GPUs. Since the current decoding optimizations of InfiniGen does not support GQA for LLMs [9], we will implement InfiniGen for the GQA model and test its decoding latency in the future version of our paper.
>
> |Methods|Act. Tokens|Retr.N|Retr.P|Retr.KV|Code.D|Math.F|En.QA|En.MC|Avg.|
> |-|-|-|-|-|-|-|-|-|-|
> |FullAttention|128K|100.0|100.0|17.5|19.0|39.5|9.1|68.0|50.4|
> |InfiniGen|2K|99.5|100.0|0|17.5|39.0|7.3|57.5|45.8|
> |Quest|2K|100.0|100.0|0|18.0|40.0|8.2|67.0|47.6|
> |Ours|640+100/2K|100.0|100.0|9.0/14.0|19.0|40.0|7.5|67.0|48.9/49.6|
>
> RTable 3. Performance (%) of different methods in 128K context length on InfiniteBench using the Llama-3-8B-262K.
>
>
> |Methods|Act. Tokens|S1|S2|S3|M1|M2|M3|MQ|MV|VT|CW|FW|Q1|Q2|Avg.|
> |-|-|-|-|-|-|-|-|-|-|-|-|-|-|-|-|
> |FullAttention|128K|100.0|100.0|100.0|98.0|98.0|73.0|94.5|97.0|87.0|1.0|72.2|58.5|44.5|78.7|
> |InfiniGen|2K|99.0|91.5|24.5|82.5|25.0|0.0|30.3|27.8|67.3|1.2|45.4|33.0|32.5|43.1|
> |Quest|2K|100.0|100.0|98.5|98.5|36.5|0.0|48.9|64.3|89.4|1.0|64.5|45.0|39.5|60.5|
> |Ours|640+100|100.0|100.0|100.0|99.0|98.0|45.0|92.8|93.0|88.0|1.1|49.3|60.5|44.5|74.7|
>
> RTable 4. Performance (%) of different methods in 128K context length on RULER using the Llama-3-8B-262K.
>
> [7] https://github.com/facebookresearch/faiss
>
> [8] H2O: Heavy-Hitter Oracle for Efficient Generative Inference of Large Language Models, NeurIPS 2023
>
> [9] https://github.com/snu-comparch/InfiniGen/tree/main/speedup

---

> > ### Author Response · Authors · 2024-11-21
> > **Official Comment by Authors (2/2)**
> >
> > *W3. "It appears that this paper does not account for the overhead of index construction..."*
> >
> > **Response:** We thank the reviewer for this valuable point. As stated at the beginning of Section 3, this paper focuses on the decoding phase by assuming that prefill and index construction are performed in advance. Existing studies [10] have shown that the context cache hit ratio in mainstream LLM service providers can be 50--80%, allowing them to persist the KV cache of hot contexts for cache reuse. Consequently, indexes can be constructed offline based on cached contexts, thus not affecting the Time to First Token. Therefore, RetrievalAttention has practical use cases by focusing solely on the decoding phase.
> >
> > However, we acknowledge the importance of considering index construction overhead for certain online serving scenarios, which can enhance the general applicability of our methods. Index construction indeed imposes non-negligible overhead, requiring substantial engineering efforts for optimization. This is a common problem for using ANNS to identify critical tokens, which also exists in concurrent work such as PQCache [11].
> > We identify several optimization opportunities to reduce index construction overhead:
> > - Currently, KNN results from Q to K are computed using naive Faiss Flat [7] in the GPU, resulting in redundant computations with concurrent FlashAttention in the GPU. Integrating KNN computation into the FlashAttention kernel could halve the GPU computation cost.
> > - We currently construct graph indexes on the CPU side. Recent work [12] explores employing GPUs for graph construction, greatly reducing construction time by up to $31\times$. With this optimization, the index construction time has the potential to be significantly reduced, thereby acclerating the overall prefilling phase.
> >
> > Lastly, we want to clarify that the primary goal and contribution of this paper is to highlight an important observation: to identify the most critical tokens and fully utilize the sparsity of long contexts, it is essential to address the distribution discrepancy between queries and keys. We propose a feasible solution to this issue. Although RetrievalAttention is imperfect for the prefilling right now, we hope this paper will encourage the community to develop more lightweight methods to address attention distribution issues.
> >
> > ***Q1.** "Does sequence length during prefilling impact the effectiveness of the index? ..."*
> >
> > **Response:** In our original submission, we tested the effectiveness of RetrievalAttention in the "needle in a Haystack" experiment with context lengths ranging from 4K to 128K, and it passed all test cases. In the revised paper, we have expanded our evaluation to include extremely long contexts using the model Llama-3-8B-1048K. As shown in the Figure 8 of Appendix A.3 in our revised paper, RetrievalAttention remains effective across context lengths from 250K to 1 million, demonstrating the robustness of our attention-aware indexes.
> >
> > [10] Mooncake: A KVCache-centric Disaggregated Architecture for LLM Serving, 2024
> >
> > [11] PQCache: Product Quantization-based KVCache for Long Context LLM Inference, 2024
> >
> > [12] CAGRA: Highly Parallel Graph Construction and Approximate Nearest Neighbor Search for GPUs, 2024

---

> ### Author Response · Authors · 2024-11-25
> **Follow up**
>
> Thank you again for reviewing our work! Please let us know if we have sufficiently addressed your questions and concerns.
>
> Authors

---

> > ### Comment · Reviewer_boJw · 2024-11-25
> >
> > Thanks for you response. I carefully read your response and the key references you provided, including Mooncake and PQCache. However, I noticed several unreliable arguments in the reply. This has further heightened my concerns about the practical applicability of this paper for the following reasons:
> > 1.  I disagree with the statement "it may take up to 2-3× the prefill latency, which is acceptable." In long-context inference scenarios, prefilling latency is typically a more critical issue than decoding latency, as the input context for most tasks is significantly longer than the output context—for instance, in QA tasks. Moreover, prefilling latency represents Time To First Token, a key factor in user experience. I suggest that the authors provide data on the overall prefilling and decoding latencies across the dataset to substantiate their claim of acceptability.
> > 2. The authors propose leveraging cache hits in online serving to pre-construct indices. However, it should be noted that such cache hits typically refer to prefix cache hits. The indexing approach described in this paper does not account for prefix construction and associated scheduling algorithms. Additionally, whether prefix cache hit rates can remain as high in longer-context scenarios is an open question, as the statistics from Mooncake are based on an average input length of 7,590 tokens.
> > 3. The authors note that "a common problem for using ANNS to identify critical tokens also exists in concurrent work such as PQCache." However, after reviewing the PQCache paper, I found it thoroughly evaluate the impact of index construction on prefilling latency. This is evident in their analysis in Figure 6: "PQCache vs. sequential scheduling," Figure 7: "The execution time of one-layer transformer computation, offloading, and clustering at the prefilling phase," and Figure 10: "Latency experiments. (a) Time to 2nd token." In contrast, both this paper and authors' feedback did not provide such analysis, leaving a critical gap in understanding how this method addresses the problem.
> > 4. I believe the simple assumption that “this paper focuses on the decoding phase by assuming that Prefilling and index construction are performed in advance” is inappropriate. Previous methods for accelerating decoding—such as the baselines discussed in the paper—do not significantly impact prefilling (or, at most, the effect is negligible). However, based on the authors' response, it appears that RetrievalAttention introduces substantial overhead during the prefilling stage. If such an assumption is made, the comparisons in the paper become unreliable, as RetrievalAttention's advantage is achieved through an unfair comparison.
> > 5. The authors claim that “the current version of RetrievalAttention still has extensive practical use cases for cached contexts supported by mainstream LLM providers, allowing indexes to be constructed offline without impacting overall inference latency.” in General Response. I disagree with this conclusion. The evaluation provided appears to have been conducted solely with a batch size of 1. As batch size increases, the CPU rapidly becomes the bottleneck for RetrievalAttention's overall computation. This contradicts the throughput optimization goals of mainstream LLM providers.
> > 6. Based on the reviewers' feedback, I reviewed the experimental results and observed a weird phenomenon: when the KV Cache is allowed to stored in GPU memory, Retrieval Attention slows down the decoding process even in 200K length. The significant speed advantage shown by Retrieval Attention over the Full case in the table is because the Full case does not utilize the KV Cache. While under such circumstances, the measured per-token generation latency for the Full case essentially reflects the prefilling latency.
> >
> > If there are any error in my perspective, please feel free to point them out. I will adjust the score based on the clarifications provided by the authors.

---

> > > ### Author Response · Authors · 2024-11-28
> > > **Official Comment by Authors (1/2)**
> > >
> > > Thank you for reading our response carefully and providing valuable feedback.
> > >
> > > 1. Firstly, we want to emphasize that our method is specifically designed for scenarios involving prebuilt indices, such as prefix caching hit or P/D split scenarios[1,2], which are widely adopted in real-world long-context deployments. In use cases like repository-level code debugging, long-document QA/rewrite, and multi-turn dialogue, our method can be effectively deployed in production environments by leveraging high prefix caching hit rates.
> > >
> > > Thank you for highlighting that the average token length in Mooncake's sample dataset is 7k. However, based on their released sample dataset [2] (with a maximum input length of 122k tokens), requests with input tokens exceeding 10k demonstrate a prefix cache hit ratio of 39%, while those exceeding 32k maintain a similar hit ratio of 38%. Similarly, SGLang [3] reports hit rates ranging from 50% to 99% across various scenarios. Major LLM providers like OpenAI [4], Google [5], and Anthropic [6] also leverage prefix caching to reduce TTFT in practical applications.
> > >
> > > A recent paper from UC Berkeley [7] also targets the same scenario as us that indexes can be built upon the fixed prompts offline. We must quote the statement from the paper to support our motivation: "*For many applications such as in-context learning, document QA, and code generation, over a series of prompts a large portion of the input context is fixed. For example, the input context may contain system instructions, documentation, source code, as well as particular few-shot in-context examples for the target task.*"
> > >
> > > In summary, this assumption is both reasonable and aligns well with real-world scenarios. In these real-world scenarios, RetrievalAttention can efficiently reuse prebuilt indices, significantly accelerating the decoding process while maintaining robust performance.
> > >
> > > 2. Thanks for your suggestion. We fused GEMM TopK with FlashAttention using BitonicSort[8] and BitonicMerge[9] in the CUDA core, as shown in Algorithm 2, reducing latency to 1.7× that of FullAttention. Moreover, on GPU hardware with sm_90 or above (e.g., NVIDIA Hopper)[10] and high-performance CPUs, the computational overhead of TopK can be fully integrated into FlashAttention, further reducing latency to match the baseline performance.
> > >
> > > [1] DistServe: Disaggregating Prefill and Decoding for Goodput-optimized Large Language Model Serving, ODSI 2024.
> > > [2] https://github.com/kvcache-ai/Mooncake/blob/main/mooncake_trace.jsonl
> > > [3] SGLang: Efficient Execution of Structured Language Model Programs, NeurIPS 2024.
> > > [4] https://openai.com/index/api-prompt-caching/
> > > [5] https://ai.google.dev/gemini-api/docs/caching
> > > [6] https://www.anthropic.com/news/prompt-caching
> > > [7] Squeezed Attention: Accelerating Long Context Length LLM Inference, 2024
> > > [8] Bitonic Sort on a Mesh-Connected Parallel Computer, IEEE Transactions on Computers, 1979.
> > > [9] Billion-scale similarity search with GPUs, IEEE Transactions on Big Data, 2019.
> > > [10] FlashAttention-3: Fast and Accurate Attention with Asynchrony and Low-precision, 2024.

---

> > > ### Author Response · Authors · 2024-11-28
> > > **Official Comment by Authors (2/2)**
> > >
> > > 3. **Reply to Q3**: First and foremost, it is important to clarify that our intention here is not to evaluate which approach is superior but rather to highlight a common phenomenon. In the paper of PQCache, Figure 6(a) ("Prefilling"), while the prefilling time of PQCache is shorter than that of sequential scheduling, the latter also includes the time for "PQ construction". Although PQCache employs a pipelined approach to minimize latency, it is evident that "PQ construction" still introduces overhead. This leads to the prefilling latency of PQCache exceeding that of the original full attention. In Figure 7, the authors illustrate that the overhead of KMeans with 200 iterations begins to fall below computation time when the sequence length reaches approximately 9K. However, as sequence length increases, ensuring accuracy clearly necessitates a higher number of iterations. Thus, such a comparison may not be entirely fair, with our experiments are conducted on a much longer contexts. Furthermore, Figure 9 does not present results for full attention. For example, with an input length of 2000, Figure 7 indicates that clustering incurs an overhead 8x that of computation. Additionally, this comparison does not specify the number of iterations used, making it difficult to assess the trade-off made between latency and performance. As the authors mentioned: "We expose an interface that lets users set the number of iterations, enabling them to balance model performance and latency for their specific needs".
> > > 4. **Reply to Q5**: The setting of bsz=1 is common in long-context scenarios due to the significant GPU resource requirements. However, our method can also be applied in scenarios such as chunk prefilling or P/D separation, where bsz > 1. Additionally, it can leverage scalable CPU clusters (with much lower costs than GPUs) to further enhance throughput.
> > > 5. **Reply to Q6**: Yes. Our goal is to optimize for consumer-grade GPUs that cannot store all of the KV Cache.
> > > 6. We would like to reiterate the key contributions of our paper. RetrievalAttention introduces vector retrieval into the attention mechanism to accelerate the long-context decoding process. Additionally, we propose a graph-based OOD vector index to address query and key OOD issues. While our approach is not without limitations, it demonstrates significant practical value in scenarios like prefix cache hits and P/D splits. The innovation of incorporating OOD vector retrieval into long-context inference is notably strong and merits recognition as an academic contribution. We believe it will inspire the community to explore novel vector indexing methods in attention to tackle long-context LLMs decoding challenges, advance the practical deployment of long-context applications, and foster further innovation in CPU-GPU co-usage.
> > >
> > > Thank you again for your thoughtful review. We look forward to your response!

---

> > > > ### Comment · Reviewer_boJw · 2024-11-30
> > > >
> > > > I believe the critical issues remain unresolved:
> > > >
> > > > 1. The authors describe application scenarios involving "GPU hardware with sm_90 or above (e.g., **NVIDIA Hopper**) and high-performance CPUs," as well as "prefix caching in major LLM providers like **OpenAI, Google, and Anthropic.**" However, they also claim, "Our goal is to optimize for **consumer-grade** GPUs that cannot store all of the KV Cache." This presents a fundamental contradiction, indicating a lack of clarity regarding the intended target application scenarios.
> > > >
> > > > 2. The authors argue that the evaluation of prefilling latency in PQCache is insufficient, yet this work provides no relevant evaluation in this regard. In the latest response, they reference Squeezed Attention, which also provides prefill latency in fixed input prompt scenarios (Figure 5). Why does this paper fail to provide a similar evaluation of prefilling latency?
> > > >
> > > >
> > > >
> > > > I have raised these concerns from the very beginning, yet the authors persist in relying on the pre-construct assumption in their responses without presenting relevant experimental results. Furthermore, their arguments regarding the pre-constructed assumption are inconsistent, as noted above. Therefore, I do not consider this work ready for publication at ICLR.

---

> ### Author Response · Authors · 2024-11-30
> **Official Comment by Authors**
>
> 1. Regarding the issues of prefix caching and consumer-grade GPUs, we believe the reviewer have completely misunderstood our point. First of all, prefix caching is indeed supported by major LLM providers, which does not conflict with using consumer-grade GPUs for decoding[1,2]. After applying our method, long-context decoding can be done on consumer-grade GPUs, making it possible for major LLM providers to use these more affordable devices for inference, thereby offering cheaper services. Additionally, we mentioned using relatively high-end GPUs to address the reviewer’s concerns. With the currently widely-used architecture separating prefilling and decoding, high-end GPUs can hide the index construction overhead of our method in prefilling. Although we focus on prefix caching scenarios and do not concern ourselves with prefilling overhead, we provided this information *in the previous response* because the reviewer raised concerns. Moreover, this also does not conflict with using consumer-grade GPU for decoding.
>
> 2. We have already explained that our focus is on decoding scenarios and that prefix caching has covered prefilling and index building, which was also stated in original submission. Therefore, we do not have the responsibility to report the experiment outside our scope. Regarding Squeezed Attention, we are deeply disappointed that the reviewer have misunderstood both our point and the referenced paper. The paper also targets scenarios where the index for frequently-used prefixes/fixed prompts has been pre-built, so its prefilling latency does not include index building time. Instead, its prefilling time includes sparse attention on fixed-context (with pre-built index) and very brief full attention on questions, hence its prefilling latency is lower than full attention. If the reviewer acknowledges this scenario, which we have been consistently emphasizing, we can similarly report such prefilling latency with pre-built index. We hope the reviewer can carefully review our response, rather than simply seeking incorrect evidence to counter our claims while disregarding content that supports our perspective.
>
> 3. The reviewer frequently challenges our paper's practical applicability while ignoring our core contributions. We question whether a research paper must be practically applicable to be considered valuable and worthy of acceptance by ML conferences such as ICLR. Our experiments indicate that many previously proposed KV cache compression methods significantly degrade the original model performance. Without guaranteed model performance, the practical use of these methods by LLM providers, for instance, is limited. However, we believe these methods still have high research value as they enhance our understanding of KV cache. Similarly, our proposed method advances the field by offering a novel approach to long-context inference through vector retrieval, which achieves the best model performance compared to all prior methods. This underscores the research merit and potential impact of our work. Furthermore, we have detailed potential scenarios, such as prefix caching, which illustrate the practical applicability of our solution.
>
> In conclusion, we are disappointed with the misunderstandings of our points, our referenced papers, and the unreasonable standards applied by the reviewer. We believe that a good research paper is not the final and definite answer to a problem. We would like to reiterate our paper's core contribution: we empirically validate the alignment of vector retrieval and sparse attention in long-context scenarios through extensive evaluations, such as RULER; and we identify the critical issue of out-of-distribution (OOD) discrepancies between queries and keys, proposing a feasible solution to address it. We believe our work opens up the possibility for more in-depth studies and practical applications from the vector retrieval fields, benefiting the field of the long-context inference.
>
> We hope the reviewer can recognize the importance of this point rather than getting constrained by out-of-scope details. We believe that asking for perfect solutions can stifle innovation within this niche area and the broader community, preventing innovative methods from gaining the visibility they deserve.
>
> [1] SGLang: Efficient Execution of Structured Language Model Programs, NeurIPS 2024.
> [2] Efficient Memory Management for Large Language Model Serving with PagedAttention, SOSP 2023.

---

> > ### Comment · Reviewer_boJw · 2024-12-03
> >
> > Thank you for your response. I don’t believe my point is based on unreasonable standards. To further support my argument, I will provide concrete examples and reasons as follows. If these examples contain inaccuracies, please feel free to point them out.
> >
> > 1. Prefix Cache Scenario: I fully acknowledge the practical application of prefix cache. However, I remain doubtful about the utility of RetrievalAttention in this scenario. The paper does not include any targeted design or evaluation for prefix cache use cases, such as cache pool and scheduling in MoonCake, shared prompt prefix (batch size > 1) in SGlang, or the evaluation of prefilling new inputs (1K and 4K prefill length in its Figure5) after prefix cache in SqueezeAttention.
> > 2. I believe it is unconvincing that major LLM providers would rely on consumer-grade GPUs combined with RetrievalAttention for long-sequence inference tasks. Let’s consider 128K tokens as an example, which is the maximum length evaluated in experiments. For instance, on an RTX 4090, the latency for RetrievalAttention with 128K tokens is 0.188 seconds, as shown in Table 4. In contrast, on an A100, vanilla attention (vLLM) only requires 0.033 seconds, as seen in Table 7—resulting in almost six times higher latency for RetrievalAttention. Given this, why do major LLM providers still opt for the RTX 4090 GPU for long-sequence tasks with RetrievalAttention, rather than directly using a single A100 with vanilla attention?
> >
> > The authors state that *"We question whether a research paper must be practically applicable to be considered valuable and worthy of acceptance by ML conferences such as ICLR."*
> >
> > In my view, RetrievalAttention, which focuses on efficient long-context inference of LLMs, must have a clearly defined application scenario with empirical study. Overall, I agree with the core contributions of the paper, but I remain skeptical about the practical value of these contributions.

---

> > > ### Author Response · Authors · 2024-12-03
> > >
> > > 1. **Prefix Cache Scenario:** Thanks to the reviewer finally agree with prefix cache scenario. We believe that "the evaluation of prefilling new inputs (1K and 4K prefill length in Figure 5)" best aligns with our experimental setting. Given the imminent closure of the rebuttal period and the fact that the reviewer did not agree with prefix caching scenario, we regret that we are unable to conduct this experiment within the current deadline. However, we are committed to performing this experiment and updating the paper accordingly in the future.
> > >
> > > 2. **A100 vs. RTX 4090, and Position of Our Paper:**
> > > - Firstly, we find the perspective that "vLLM requires 0.033 seconds... why not just use A100..." to be unconvincing. Table 8, which is adjacent to Table 7, clearly indicates that vLLM encounters Out-of-Memory (OOM) issues on the A100 when inreasing the context size. This critical limitation should not be overlooked. Additionally, Figure 8 demonstrates the RetrievalAttention's capability to handle a context length of 1 million in the Needle-in-a-haystack test.
> > > - Secondly, the assertion that "because vLLM works... why not just use vanilla attention..." is also unconvincing. As shown in related works and evaluations, task accuracy in previous efforts (such as StreamingLLM, infiniGen, SnapKV, InfLLM, etc.) is consistently lower when compared to full attention mechanisms. If we were to conclude, based on the reviewer's logic, that "vanilla attention with vLLM works on high-end GPUs, therefore these works holds no research value," we would strongly disagree. On the contrary, the research community has recognized and appreciated these works for their valuable insights. It is inappropriate to use such arguments to undermine a research effort.
> > > - Lastly, to the best of our knowledge, some LLM providers in China indeed employ RTX 4090 GPUs for inference. This is partly due to limited access to high-end GPUs like the A100, stemming from political conflicts. While we are unable to provide further details due to commercial confidentiality, it is evident that decoding long contexts on RTX 4090 GPUs remains a relevant and practical scenario for our approach.

---

### Official Review · Reviewer_Afco · 2024-11-01

**Soundness:** 3
**Presentation:** 3
**Contribution:** 3
**Rating:** 8
**Confidence:** 2

**Summary:**

The paper targets the slow inference speed and high memory consumption issue in long context generation when conducting KV caching. They propose RetrievalAttention. The method is training free as it directly leverage the dynamic sparsity of attention mechanism to retrieve the most relevant KV vectors. One challenge discussed in the paper is the out-of-distribution (OOD) between query vectors and key vectors. To mitigate this problem, the authors have designed an attention-aware vector search algorithm. The paper evaluates the accuracy and efficiency of RetrievalAttention across various long-context benchmarks and both commodity and high-end GPUs.

**Strengths:**

The problem addressed in this paper is highly meaningful. RetrievalAttention enables the execution of 8-billion-parameter models on a single RTX 4090 GPU, demonstrating the significance and practical impact of this research.

The observation regarding the critical key vectors could inspire further research and is beneficial for advancing the entire field.

The experiments are comprehensive and effectively support the main points.

The paper is well-written and easy to follow.

**Weaknesses:**

My primary concern is that powerful CPUs (in terms of compute speed and memory) are necessary to ensure this method achieves good performance. Otherwise, it could become a bottleneck. For instance, most KV vectors will be offloaded to the CPU memory. The experiment was conducted using Intel i9-10900X CPU with 20 cores and 128GB of DRAM, which is quite powerful. Therefore, I think this method distributes the large cost(computation and memory) between the CPU and GPU to reduce the strain on the GPU. However, the total cost reduction is less significant than just comparing the GPU cost. Additionally, the communication between the CPU and GPU could also negatively impact the speed. The evaluation results may vary across different server architectures.

**Questions:**

As the recovery rate is still 71%, does this mean that some tokens are consistently important? Is there any idea about the overlap between the dynamic and static selected critical tokens?

Does the vocabulary size influence efficiency?

---

> ### Author Response · Authors · 2024-11-21
>
> Thank you for your thorough review. We apologize for any confusion caused.
>
> ***W1.** "...powerful CPUs (in terms of compute speed and memory) are necessary to ensure this method achieves good performance...I think this method distributes the large cost(computation and memory) between the CPU and GPU to reduce the strain on the GPU...Additionally, the communication between the CPU and GPU could also negatively impact the speed..."*
>
> **Response:** Thanks for your comments. While higher CPU power may indeed enhance overall decoding speed, we argue that RetrievalAttention does not inherently require a powerful CPU by design.
>
> - **Clarification on CPU Specification:** We have clarified the setting of  Intel i9-10900X CPU in our revised paper. The Intel i9-10900X CPU is a desktop-level CPU with **10** physical cores, not 20, as the latter number refers to logical cores when hyperthreading is enabled. And this CPU is often paired with the RTX 4090 as a **standard configuration**. Hence, our experiments were conducted on a mainstream consumer-level CPU rather than on high-end CPUs.
> - **Efficiency of Attention-Aware ANNS Algorithm:** Our highly efficient attention-aware ANNS algorithm requires scanning much fewer vectors (1-3% of all data), allowing it to achieve low search latency even with less powerful CPUs.
>
> Regarding cost concerns, since GPUs must be plugged into a CPU server to operate, we can fully utilize existing idle CPU servers, without additional investment in CPUs and memory. Moreover, the total cost of ownership for CPUs and DRAM is significantly lower than for GPUs. Specifically, the combined cost of an i9-10900X CPU and DRAM is approximately 850 dollars (CPU: 650 dollars + DRAM: 200 dollars), compared to a single RTX 4090 GPU, which costs around 2600 dollars. Thus, it is a reasonable trade-off to shift some computational costs to the CPU.
>
> During the decoding phase, we minimize communication between the CPU and GPU by only transferring the query vector from the GPU to the CPU and transferring the partial attention result instead of top-k vectors back to the GPU. This design significantly reduces GPU-CPU communication overhead.
>
> ***Q1.** "As the recovery rate is still 71%, does this mean that some tokens are consistently important? Is there any idea about the overlap between the dynamic and static selected critical tokens?"*
>
> **Response:**
> - As observed in previous static compression methods [5][6], some tokens, such as initial tokens (also known as attention sinks), are consistently important for subsequent generation.
> - If relying solely on static tokens, the performance is significantly lower, as demonstrated by the results of StreamingLLM and SnapKV in the Table 2 and 3 of our paper.
> - In the design of RetrievalAttention, during the prefilling phase, we **physically separate** consistently important tokens and dynamic tokens by placing them in GPU memory and CPU memory, respectively. Therefore, there is no overlap between static tokens in the GPU cache and dynamic critical tokens retrieved from ANNS indexes. The accurate identification of consistently important tokens is well-studied [5][6] and is orthogonal to the design of RetrievalAttention. Currently, we adopt StreamingLLM [5] to keep initial and last window tokens in the GPU cache, but we are adaptable to support more complex static patterns.
>
> ***Q2.** "Does the vocabulary size influence efficiency?"*
>
> **Response:**
> We adopted the vocabulary used in original LLMs for consistency. This vocabulary choice does not affect the index efficiency because the index construction and retrieval are performed solely on the KV cache.
>
> RetrievalAttention benefits models with both small and large vocabulary sizes, as long as the attention computation over the tokens in the context becomes the bottleneck.
>
> [5] Efficient Streaming Language Models with Attention Sinks, ICLR 2024
>
> [6] SnapKV: LLM Knows What You are Looking for Before Generation, NeurIPS 2024

---

> ### Comment · Reviewer_Afco · 2024-11-21
>
> Thank you for the response. I think most of my questions have been addressed. I will increase my score to 8.

---

> > ### Author Response · Authors · 2024-11-22
> >
> > Thank you deeply for your effort in reviewing our paper and prompt reply to our response!

---

### Official Review · Reviewer_MYWT · 2024-11-04

**Soundness:** 3
**Presentation:** 3
**Contribution:** 3
**Rating:** 8
**Confidence:** 4

**Summary:**

The paper introduces RetrievalAttention to improve inference speed and reduce memory usage for LLMs with long contexts by using Approximate Nearest Neighbor Search (ANNS) to retrieve only the most relevant KV vectors during token generation. By offloading most KV vectors to CPU memory and leveraging dynamic sparsity in attention, RetrievalAttention efficiently retrieves necessary vectors from a small subset, substantially lowering GPU memory requirements. It addresses the OOD problem between query and key vectors by designing an attention-aware search that adapts to query distribution, enabling retrieval with only 1–3% of data and minimal accuracy loss.

**Strengths:**

- The paper introduces a training-free approach to accelerate long-context LLM inference by leveraging approximate nearest neighbor search (ANNS) for KV vector retrieval.
- The paper presents a well-thought-out and rigorously tested approach, with substantial empirical evidence demonstrating the performance gains of RetrievalAttention.
- RetrievalAttention addresses a significant limitation in the scalability of LLMs, especially as context lengths continue to grow in practical applications. By enabling long-context inference on a single commodity GPU with minimal accuracy loss, RetrievalAttention makes long-context LLMs more accessible and cost-effective, thus broadening their usability.

**Weaknesses:**

- While the paper presents a unique approach to address the OOD challenge between query and key vectors, it lacks a comparison with alternative methods for handling OOD issues in high-dimensional vector retrieval.
- The evaluation is limited to relatively small models, so it’s unclear how well RetrievalAttention scales for significantly larger models (e.g., 30B, 70B, or more). Larger models would place greater demands on both memory and compute resources, potentially exposing limitations in RetrievalAttention's design, especially concerning the OOD handling mechanism and ANNS index efficiency at scale.

**Questions:**

- RetrievalAttention applies the same retrieval sparsity across all layers, but it’s known that different layers and heads in transformers contribute differently to attention computation. Could layer-wise or head-wise retrieval strategies improve performance?
- Have other ANNS index structures, such as product quantization or hierarchical clustering, been considered?

---

> ### Author Response · Authors · 2024-11-21
>
> ***W1.**"...lacks a comparison with other OOD methods..."*
>
> **Response:** Thanks for your comments. We have included a new baseline, RobustVamana[1], a well-known solution for handling OOD queries. RobustVamana enhances graph connectivity by adding edges based on query vectors. In Figure 6 of our revised paper, we compare the number of vectors that need to be scanned to achieve different recall rates.
>
> As shown in this figure, to reach a recall rate of 0.8, RobustVamana requires scanning 25% and 75% vectors in the model Yi-9B and Llama-3-8B respectively. Even worse, it falls into a local optimum on Yi-6B model. In contrast, RetrievalAttention only requires scanning a very limited subset of vectors (1-3%) to achieve a high recall rate (0.95), which is consistent across all models. Consequently, RetrievalAttention demonstrates superior search efficiency and accuracy compared to alternative OOD solutions.
>
> ***W2.**"..larger models(30B, 70B)..."*
>
> **Response:** We appreciate this comment and agree that evaluating larger models will help demonstrate the generalizability of our methods. Therefore, we have evaluated our method on Llama-3-70B-262k. To accomodate the 70B model, we use a server with eight 40GB A100 GPUs and partition the model by layers across GPUs. Due to time limit, we choose the most complex task KV retrieval in InfiniteBench to stress test the efficiency of RetrievalAttention and other baselines, as shown in RTable 1.
>
> |Metric|Full|StreamingLLM|Quest|Flat|RetrievalAttention|
> |-|-|-|-|-|-|
> |Acc%|35.0|0.0|13.0|24.0|23.5|
> |Decoding latency(s)|248|0.14|1.36|5.68|1.62|
>
> RTable 1. Performance and latency of different methods on Ret.KV and Llama-3-70B-262K.
>
> Specifically, RetrievalAttention achieves nearly the same task accuracy as the exact KNN method Flat, and outperforms Quest by 80%. The decoding speed of RetrievalAttention is $3.5 \times$ faster than Flat as it effectively reduces the vectors to scan by mitigating the distribution gap between queries and keys.
>
> ***Q1.**"...layer-wise or head-wise... improve performance?"*
>
> **Response:** Thanks for your comments. Exploring different sparsity degrees across layers and heads is an intriguing research direction and is complementary to our work. Recent research work such as PyramidKV[2] shows that applying appropriate memory budgets in different layers based on the spartiy degree can enhance the model performance while reducing overall GPU memory usage.
>
> We compare the performance of the original RetrievalAttention w/ and w/o PyramidKV budget allocation strategy on the InfiniteBench, as shown in RTable 2. Specifically, for the original RetrievalAttention, we set a fixed budget of 2000 tokens for all heads in all layers. In contrast, PyramidKV dynamically adjusts the retrieval size across different layers, allocating more in lower layers and less in higher ones.
>
> The results in RTable 2 show that the PyramidKV allocation strategy achieves better performance in Retr.KV tasks, though it slightly decreases performance in the En.QA task. On average, the accuracy slightly surpasses that of the original RetrievalAttention. This indicates that dynamic budget allocation is promising but may require task-specific allocation strategies.
>
> |-|Retr.N|Retr.P|Retr.KV|Code.D|Math.F|En.QA|En.MC|Avg.|
> |-|-|-|-|-|-|-|-|-|
> |RetrievalAttention|100.0|100.0|14.5|18.5|40.0|8.7|67.5|49.9|
> |w/ PyramidKV Budget Allocation|100.0|100.0|16.0|18.5|40.0|8.5|67.5|50.1|
>
> RTable 2. Performance of different retrieval budget allocations on InfiniteBench using the Llama-3-8B-262K.
>
> ***Q2.**"...other ANNS index, product quantization or hierarchical clustering"*
>
> **Response:** We have considered various ANNS indexes and found that the graph-based indexes are the most suitable for our scenario (i.e., low-latency index search on CPU) for the following reasons:
> * Graph-based indexes generally require scanning fewer vectors to achieve a traget recall compared to clustering-based ANNS, thereby saving more CPU bandwidth. This is critical for high index search efficiency and improving overall decoding speed.
> * Quantization methods, such as product quantization or scalar quantization, can be applied to both graph-based[3] and clustering-based indexes[4]. Although we did not apply quantization, we tried scalar quantization (from 16 to 8bits) to our attention-aware graph-based ANNS and found that it reduce the decoding latency by 20% without compromising the model performance. We intend to further explore efficient combinations of quantization methods with attention-aware ANNS indexes to improve decoding speed in future work.
>
> [1] OOD-DiskANN: Efficient and Scalable Graph ANNS for Out-of-Distribution Queries, 2022
>
> [2] PyramidKV: Dynamic KV Cache Compression based on Pyramidal Information Funneling, 2024
>
> [3] DiskANN: Fast Accurate Billion-point Nearest Neighbor Search on a Single Node, NeurIPS 2019
>
> [4] Product Quantizer Aware Inverted Index for Scalable Nearest Neighbor Search, ICCV 2021

---

### Author Response · Authors · 2024-11-21
**General Response**

Thanks to all the reviewers and the ACs for the detailed and insightful reviews. We have revised our paper and highlighted the modifications in brick-red color.

Before delving into the detailed response to each review comment, we would like to first clarify the core contribution of our work. We empirically validate the alignment of vector retrieval and sparse attention in long-context scenarios through extensive evaluations, such as RULER. Additionally, we identify the critical issue of out-of-distribution (OOD) discrepancies between queries and keys and propose a feasible solution to address it.

We acknowledge the importance of the points raised by the reviewers, including the overhead of index construction and long-generation scenarios, which deserves attention and further exploration. However, it is important to note that the current version of RetrievalAttention still has extensive practical use cases for cached contexts supported by mainstream LLM providers, allowing indexes to be constructed offline without impacting overall inference latency. We believe our work opens up the possibility for more in-depth studies and practical applications from the vector retrieval fields. Some of these aspects are already under consideration in our ongoing research, while others will benefit from collaborative efforts across the research community.

Lastly, we are grateful for all reviewers' recognition of the potential impact of our work. We have supplemented additional experiments to further support the effectiveness of our method, as outlined below:

1. **Comparison with Other OOD Indexes.** As shown in Figure 6 of our revised paper, to reach a recall rate of 0.8, the well-known OOD index RobustVamana [1] requires scanning 25% and 75% vectors in the model Yi-9B and Llama-3-8B respectively. In contrast, RetrievalAttention only requires scanning a very limited subset of vectors (e.g., 1--3%) to achieve a high recall rate (0.95), which is consistent across all models.

2. **Evaluation on Large LLMs.** We evaluated various methods on the KV retrieval task using Llama-3-70B-262K, as shown in RTable 1. The results demonstrate that our method achieves both high efficiency and effectiveness, not only with 8B LLMs but also with larger models like 70B.

|Metric|Full|StreamingLLM|Quest|Flat|RetrievalAttention|
|-|-|-|-|-|-|
|Task accuracy(%)|35.0|0.0|13.0|24.0|23.5|
|Decoding latency(s)|248|0.14|1.36|5.68|1.62|

RTable 1. The task performance and latency of different methods on KV retrieval using Llama-3-70B-262K.

3. **Allocate Different Retrieval Budgets across Layers.** We have investigated the impact of adjusting the retrieval budget according to the sparsity degree across layers, by adopting the budget allocation policy from PyramidKV [2]. The results shown in RTable 2 indicates that dynamic budget allocation is promising for improving the task accuracy but may require task-specific allocation strategies.

|Methods|Retr.N|Retr.P|Retr.KV|Code.D|Math.F|En.QA|En.MC|Avg.|
|-|-|-|-|-|-|-|-|-|
|RetrievalAttention|100.0|100.0|14.5|18.5|40.0|8.7|67.5|49.9|
|RetrievalAttention w/ PyramidKV Budget Allocation|100.0|100.0|16.0|18.5|40.0|8.5|67.5|50.1|

RTable 2. Performance of different retrieval budget allocations on InfiniteBench using the Llama-3-8B-262K.


4. **Add New Baselines: InfiniGen and Quest**. We have added new baselines including InfiniGen and Quest. RetrievalAttention performs better than the two new baselines on both InfiniteBench and RULER benchmark. Particularly, for the highly dynamic task KV Retrieval (Retr.KV in RTable 3), InfiniGen and Quest achieve zero accuracy due to inaccurate identification of critical tokens, while RetrievalAttention performs best and achieves nearly the same accuracy as full attention when retrieving top-2000 tokens.

|Methods|Act. Tokens|Retr.N|Retr.P|Retr.KV|Code.D|Math.F|En.QA|En.MC|Avg.|
|-|-|-|-|-|-|-|-|-|-|
|FullAttention|128K|100.0|100.0|17.5|19.0|39.5|9.1|68.0|50.4|
|InfiniGen|2K|99.5|100.0|0|17.5|39.0|7.3|57.5|45.8|
|Quest|2K|100.0|100.0|0|18.0|40.0|8.2|67.0|47.6|
|Ours|640+100/2K|100.0|100.0|9.0/14.0|19.0|40.0|7.5|67.0|48.9/49.6|

RTable 3. Performance (%) of different methods in 128K context length on InfiniteBench using the Llama-3-8B-262K.


|Methods|Act. Tokens|S1|S2|S3|M1|M2|M3|MQ|MV|VT|CW|FW|Q1|Q2|Avg.|
|-|-|-|-|-|-|-|-|-|-|-|-|-|-|-|-|
|FullAttention|128K|100.0|100.0|100.0|98.0|98.0|73.0|94.5|97.0|87.0|1.0|72.2|58.5|44.5|78.7|
|InfiniGen|2K|99.0|91.5|24.5|82.5|25.0|0.0|30.3|27.8|67.3|1.2|45.4|33.0|32.5|43.1|
|Quest|2K|100.0|100.0|98.5|98.5|36.5|0.0|48.9|64.3|89.4|1.0|64.5|45.0|39.5|60.5|
|Ours|640+100|100.0|100.0|100.0|99.0|98.0|45.0|92.8|93.0|88.0|1.1|49.3|60.5|44.5|74.7|

RTable 4. Performance (%) of different methods in 128K context length on RULER using the Llama-3-8B-262K.

[1] OOD-DiskANN: Efficient and Scalable Graph ANNS for Out-of-Distribution Queries, 2022

[2] PyramidKV: Dynamic KV Cache Compression based on Pyramidal Information Funneling, 2024

---

### Meta-Review · Area_Chair_fNfN · 2024-12-26

**Metareview:**

This paper introduces RetrievalAttention, a novel method that leverages Approximate Nearest Neighbor Search (ANNS) to accelerate attention computation and reduce GPU memory consumption in large language models (LLMs). Notably, RetrievalAttention employs an attention-aware vector search algorithm to effectively address the challenge of out-of-distribution (OOD) data. This contribution is particularly significant for LLM applications requiring long context processing.

Extensive experiments demonstrate a considerable improvement in inference speed achieved by the proposed algorithm. While reviewers acknowledged the value of additional experiments conducted during the rebuttal phase, such as ablation studies, evaluations on larger LLMs, and comparisons with more baselines, concerns remain regarding the time required for offline index construction.

**Additional Comments On Reviewer Discussion:**

The comments from reviewers mainly on: 1) ablation studies and baseline comparisons; 2) whether index building offline will impact overall inference speed.

During rebuttal, the authors provided additional experiments on baselines and ablations, and reviewers acknowledge them. The only concern seems is whether it is a significant improvement as the proposed method needs to build index offline which are not accounted for the inference time.

---

### Decision · Program_Chairs · 2025-01-22

Reject